# INDUCTIVE REPRESENTATION LEARNING IN TEMPORAL NETWORKS VIA CAUSAL ANONYMOUS WALKS

**Yanbang Wang**[1]*, **Yen-Yu Chang**[2], **Yunyu Liu**[3], **Jure Leskovec**[1], **Pan Li**[1,3]
[1]Department of Computer Science, [2]Electrical Engineering, Stanford University
[3]Department of Computer Science, Purdue University
`{ywangdr,jure}@cs.stanford.edu,yenyu@stanford.edu`
`{liu3154,panli}@purdue.edu`

## ABSTRACT

Temporal networks serve as abstractions of many real-world dynamic systems. These networks typically evolve according to certain laws, such as the law of triadic closure, which is universal in social networks. Inductive representation learning of temporal networks should be able to capture such laws and further be applied to systems that follow the same laws but have not been unseen during the training stage. Previous works in this area depend on either network node identities or rich edge attributes and typically fail to extract these laws. Here, we propose *Causal Anonymous Walks (CAWs)* to inductively represent a temporal network. CAWs are extracted by temporal random walks and work as automatic retrieval of temporal network motifs to represent network dynamics while avoiding the time-consuming selection and counting of those motifs. CAWs adopt a novel anonymization strategy that replaces node identities with the hitting counts of the nodes based on a set of sampled walks to keep the method inductive, and simultaneously establish the correlation between motifs. We further propose a neural-network model CAW-N to encode CAWs, and pair it with a CAW sampling strategy with constant memory and time cost to support online training and inference. CAW-N is evaluated to predict links over 6 real temporal networks and uniformly outperforms previous SOTA methods by averaged 15% AUC gain in the inductive setting. CAW-N also outperforms previous methods in 5 out of the 6 networks in the transductive setting.

## 1 INTRODUCTION

Temporal networks consider dynamically interacting elements as nodes, interactions as temporal links, with labels of when those interactions happen. Such temporal networks provide abstractions to study many real-world dynamic systems (Holme & Saramäki, 2012). Researchers have investigated temporal networks in recent several decades and concluded many insightful laws that essentially reflect how these real-world systems evolve over time (Kovanen et al., 2011; Benson et al., 2016; Paranjape et al., 2017; Zitnik et al., 2019). For example, the law of triadic closure in social networks, describing that two nodes with common neighbors tend to have a mutual interaction later, reflects how people establish social connections (Simmel, 1950). Later, a more elaborate law on the correlation between the interaction frequency between two individuals and the degree that they share social connections, further got demonstrated (Granovetter, 1973; Toivonen et al., 2007). Feedforward control loops that consist of a direct interaction (from node $w$ to node $u$) and an indirect interaction (from $w$ through another node $v$ to $u$), also work as a law in the modulation of gene regulatory systems (Mangan & Alon, 2003) and also as the control principles of many engineering systems (Gorochowski et al., 2018). Although research on temporal networks has achieved the above success, it can hardly be generalized to study more complicated laws: Researchers have to investigate an exponentially increasing number of patterns when incorporating more interacting elements let alone their time-evolving aspects.

Recently, representation learning, via learning vector representations of data based on neural networks, has offered unprecedented possibilities to extract, albeit implicitly, more complex structural patterns (Hamilton et al., 2017b; Battaglia et al., 2018). However, as opposed to the study on static networks, representation learning of temporal networks is far from mature. Two challenges on temporal networks have been frequently discussed. First, the entanglement of structural and temporal

---

*Project website with code and data: `http://snap.stanford.edu/caw/`

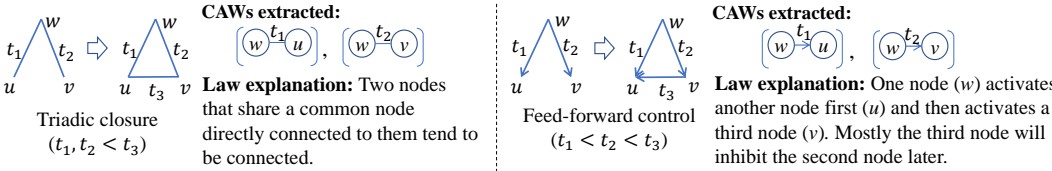

Figure 1: Triadic closure and feed-forward loops: Causal anonymous walks (CAW) capture the laws.

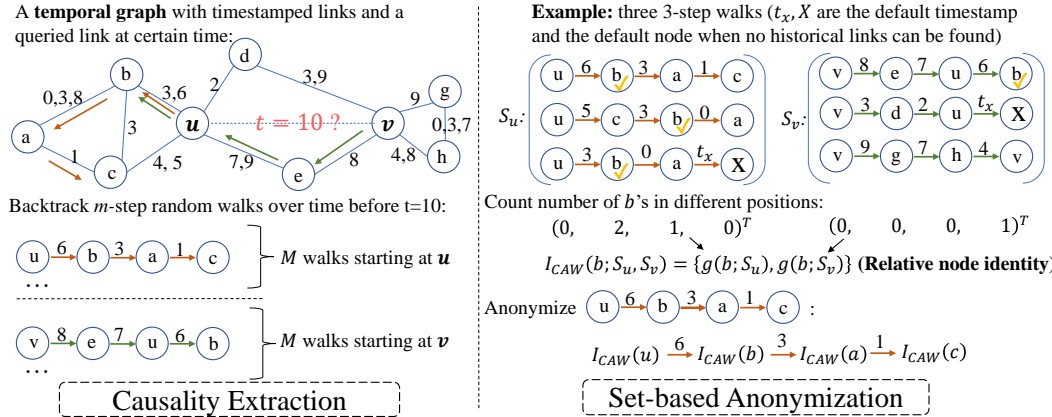

Figure 2: Causal anonymous walks (CAW): causality extraction and set-based anonymization.

patterns required an elegant model to digest the two-side information. Second, the model scalability becomes more crucial over temporal networks as new arriving links need to be processed timely while a huge link set due to the repetitive links between two nodes needs to be digested simultaneously.

In contrast to the above two challenges, another challenge, the inductive capability of the temporal-network representation, is often ignored. However, it is equally important if not more, as the inductive capability indicates whether the models indeed capture the dynamic laws of the systems and can be further generalized to the system that share the same laws but have not been used to train these models. These laws may only depend on structures such as the triadic closure or feed-forward control loops as aforementioned. These laws may also correlate with node attributes, such as interactions between people affected by their gender and age (Kovanen et al., 2013). But in both cases, the laws should be independent from network node identities. Although previous works tend to learn inductive models by removing node identities (Trivedi et al., 2019; Xu et al., 2020), they run into other issues to inductively represent the dynamic laws, for which we leave more detailed discussion in Sec. 2.

Here we propose *Causal Anonymous Walks (CAW)* for modeling temporal networks. Our idea for inductive learning is inspired by the recent investigation on temporal network motifs that correspond to connected subgraphs with links that appear within a restricted time range (Kovanen et al., 2011; Paranjape et al., 2017). Temporal network motifs essentially reflect network dynamics: Both triadic closure and feed-forward control can be viewed as temporal network motifs evolving (Fig. 1); An inductive model should predict the 3rd link in both cases when it captures the correlation of these two links as they share a common node, while the model is agnostic to the node identities of these motifs.

Our CAW model has two important properties (Fig. 2): (1) Causality extraction — a CAW starts from a link of interest and backtracks several adjacent links over time to encode the underlying causality of network dynamics. Each walk essentially gives a temporal network motif; (2) Set-based anonymization — CAWs remove the node identities over the walks to guarantee inductive learning while encoding *relative node identities* based on the counts that they appear at a certain position according to a set of sampled walks. Relative node identities guarantee that the structures of motifs and their correlations are still kept after removing node identities. To predict temporal links between two nodes of interest, we propose a model CAW-Network (CAW-N) that samples a few CAWs related to the two nodes of interest, encodes and aggregates these CAWs via RNNs (Rumelhart et al., 1986) and set-pooling respectively to make the prediction.

Experiments show that CAW-N is extremely effective. CAW-N does not need to enumerate the types of motifs and count their numbers that have been used as features to predict network dynamics (Lahiri & Berger-Wolf, 2007; Rahman & Al Hasan, 2016; Rossi et al., 2019; AbuOda et al., 2019; Li & Milenkovic, 2017), which significantly saves feature-engineering effort. CAW-N also keeps all fine-grained temporal information along the walks that may be removed by directly counting motifs (Ahmed et al., 2015; Paranjape et al., 2017). CAWs share a similar idea as anonymous walks (AW) (Micali & Zhu, 2016) to remove node identities. However, AWs have only been used for entire static graph embedding (Ivanov & Burnaev, 2018) and are not directedly applied to represent temporal networks: AWs cannot capture causality; AWs get anonymized based on each single walk and hence lose the correlation between network motifs. In contrast, CAWs capture all the information, temporal, structural, motif-correlation that are needed, to represent temporal networks.

We conclude our contributions in three-folds: (1) A novel approach to represent temporal network CAW-N is proposed, which leverages CAWs to encode temporal network motifs to capture network dynamics while keeping fully inductive. CAW-N is evaluated to predict links over 6 real-world temporal networks. CAW-N outperforms all SOTA methods by about 15% averaged over 6 networks in the inductive setting and also significantly beat all SOTA methods over 5 networks in the transductive setting; (2) CAW-N significantly decreases the feature-engineering effort in traditional motif selection and counting approaches and keeps fine-grained temporal information; (3) CAW-N is paired with a CAW sampling method with constant memory and time cost, which conduces to online learning.

## 2 RELATED WORK

Prior work on representation learning of temporal networks preprocesses the networks by simply aggregating the sequence of links within consecutive time windows into network snapshots, and use graph neural networks (GNN) (Scarselli et al., 2008; Kipf & Welling, 2017) and RNNs or transformer networks (Vaswani et al., 2017) to encode structural patterns and temporal patterns respectively (Pareja et al., 2020; Manessi et al., 2020; Goyal et al., 2020; Hajiramezanali et al., 2019; Sankar et al., 2020). The main drawback of these approaches is that they need to predetermine a time granularity for link aggregation, which is hard to learn structural dynamics in different time scales. Therefore, approaches that work on link streams directly have been recently proposed (Trivedi et al., 2017; 2019; Kumar et al., 2019; Xu et al., 2020). Know-E (Trivedi et al., 2017), DyRep (Trivedi et al., 2019) and JODIE (Kumar et al., 2019) use RNNs to propagate messages across interactions to update node representations. Know-E, JODIE consider message exchanges between two directly interacted nodes while DyRep considers an additional hop of interactions. Therefore, DyRep gives a more expressive model at a cost of high complexity. TGAT (Xu et al., 2020) in contrast mimics GraphSAGE (Hamilton et al., 2017a) and GAT (Veličković et al., 2018) to propagate messages in a GNN-like way from sampled historical neighbors of a node of interest. TGAT's sampling strategy requires to store all historical neighbors, which is unscalable for online learning. Our CAW-N directly works on link streams and only requires to memorize constant many most recent links for each node.

Most of the above models are not inductive because they associate each node with an one-hot identity (or the corresponding row of the adjacency matrix, or a free-trained vector) (Li et al., 2018; Chen et al., 2019; Kumar et al., 2019; Hajiramezanali et al., 2019; Sankar et al., 2020; Manessi et al., 2020; Goyal et al., 2020).

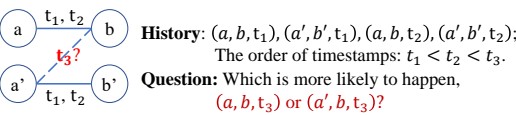

**History**: $(a, b, t_1), (a', b', t_1), (a, b, t_2), (a', b', t_2)$;
The order of timestamps: $t_1 < t_2 < t_3$.
**Question:** Which is more likely to happen, $(a, b, t_3)$ or $(a', b, t_3)$?

Figure 3: Ambiguity due to removing node identities in TGAT (Xu et al., 2020) ($t_1 < t_2 < t_3$).

TGAT (Xu et al., 2020) claimed to be inductive by removing node identities and just encoding link timestamps and attributes. However, TGAT was only evaluated over networks with rich link attributes, where the structural dynamics is not captured essentially: If we focus on structural dynamics only, it is easy to show a case when TGAT confuses node representations and will fail: Suppose in the history, two node pairs $\{a, b\}$ and $\{a', b'\}$ only interact within each pair but share the timestamps (Fig. 3). Intuitively, a proper model should predict that future links still appear within each pair. However, TGAT cannot distinguish $a$ v.s. $a'$, and $b$ v.s. $b'$, which leads to incorrect prediction. Note that GraphSAGE (Hamilton et al., 2017a) and GAT (Veličković et al., 2018) also share the similar issue when representing static networks for link prediction (Zhang et al., 2020; Srinivasan & Ribeiro, 2019). DyRep (Trivedi et al., 2019) is able to relieve such ambiguity by merging node representations with their neighbors' via RNNs. However, when DyRep runs over a new network, it frequently encounters node representations unseen during its training and will fail to make correct prediction.

**Algorithm 1:** Temporal Walk Extraction $(\mathcal{E}, \alpha, M, m, w_0, t_0)$

1   Initialize $M$ walks: $W_i \leftarrow ((w_0, t_0)), 1 \leq i \leq M$ ;
2   **for** $j$ *from* 1 *to* $m$ **do**
3     **for** $i$ *from* 1 *to* $M$ **do**
4       $(w_{\text{p}}, t_{\text{p}}) \leftarrow$ the last (node, time) pair in $W_i$;
5       Sample one $(e, t) \in \mathcal{E}_{w_{\text{p}}, t_{\text{p}}}$ with prob. $\propto \exp(\alpha(t - t_{\text{p}}))$
        Denote $e = \{w', w\}$ and then $W_i \leftarrow W_i \oplus (w', t)$;

6   Return $\{W_i | 1 \leq i \leq M\}$;

**The rule:** "*one node (e.g. u) interacts with other nodes only if another node interacts with this node at least twice.*"

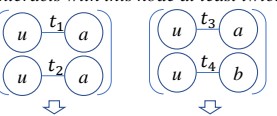

$u$ takes action     $u$ NOT takes action

Figure 4: The correlation between walks needs to be captured to learn this law.

Our CAW-N removes node identities and leverages relative node identities to avoid the issue in Fig. 3. Detailed explanations are given in Sec.4.2.

Network-embedding approaches may also be applied to temporal networks (Zhou et al., 2018; Du et al., 2018; Mahdavi et al., 2018; Singer et al., 2019; Nguyen et al., 2018). However, they directly assign each node with a learnable vector. Therefore, they are not inductive and cannot digest attributes.

## 3   PROBLEM FORMULATION AND NOTATIONS

**Problem Formulation**. A temporal network can be represented as a sequence of links that come in over time, *i.e.* $\mathcal{E} = \{(e_1, t_1), (e_2, t_2), ...\}$ where $e_i$ is a link and $t_i$ is the timestamp showing when $e_i$ arrives. Each link $e_i$ corresponds to a dyadic event between two nodes $\{v_i, u_i\}$. For simplicity, we first assume those links to be undirected and without attributes while later we discuss how to generalized our method to directed attributed networks. The sequence of links encodes network dynamics. Therefore, the capability of a model for representation learning of temporal networks is typically evaluated by how accurately it may predict future links based on the historical links (Sarkar et al., 2012). In this work, we also use link prediction as the metric. Note that we care not only the link prediction between the nodes that have been seen during the training. We also expect the models to predict links between the nodes that has never been seen as the inductive evaluation.

**Notations**. We define $\mathcal{E}_{v,t} = \{(e, t') \in \mathcal{E} | t' < t, v \in e\}$ to include the links attached to a node $v$ before certain time $t$. A walk $W$ (reverse over time) on temporal networks can be represented as

$$W = ((w_0, t_0), (w_1, t_1), ..., (w_m, t_m)), \ t_0 > t_1 > \cdots > t_m, \ (\{w_{i-1}, w_i\}, t_i) \in \mathcal{E} \text{ for all } i. \quad (1)$$

We use $W[i]$ to denote the $i$th node-time pair, and $W[i][0]$ and $W[i][1]$ to denote the corresponding node and time in $W[i]$ correspondingly. Later, we also use $\oplus$ as vector concatenation.

*Temporal network motifs* are defined as connected subgraphs that consist of links appearing within a restricted time range (Kovanen et al., 2011). Based this definition, each walk defined in Eq. 1 naturally corresponds to a temporal network motif as long as $(t_1 - t_m)$ is in the time range.

## 4   PROPOSED METHOD: CAUSAL ANONYMOUS WALK-NETWORK

### 4.1   PRELIMINARIES: ANONYMOUS WALK AND TEMPORAL NETWORK MOTIF

*Anonymous walks* were first considered by Micali & Zhu (2016) to study the reconstruction of a Markov process from the records without sharing a common "name space". AWs can be directly rephrased in the network context. Specifically, an AW starts from a node, performs random walks over the graph to collect a walk of nodes, e.g. $(v_1, v_2, ..., v_m)$. AW has an important anonymization step to replace the node identities by the orders of their appearance in each walk, which we term relative node identities in AW and define it as

$$I_{AW}(w; W) \triangleq |\{v_0, v_1, ..., v_{k^*}\}|, \text{ where } k^* \text{ is the smallest index s.t. } v_{k^*} = w. \quad (2)$$

Note that the set operation above removes duplicated elements. Although AW removes node identities, those nodes are still distinguishable within this walk. Therefore, an AW can be viewed as a network motif while the information on which specific nodes form this motif is removed. Examples of AWs are shown as follows. While these are two different walks, they may be mapped to the same AW when node identities get removed.

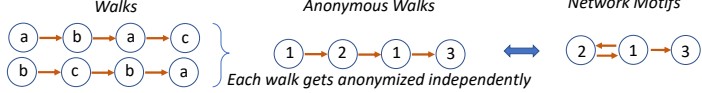

## 4.2 CAUSAL ANONYMOUS WALK

We propose CAW that shares the high-level concept with AW to remove the original node identities. However, CAW has a different causal sampling strategy and a novel set-based approach for node anonymization, which are specifically designed to encode temporal network dynamics (Fig. 2).

**Causality Extraction**. Alg. 1 shows our causal sampling: We sample connected links by backtracking over time to extract the underlying causality of network dynamics. More recent links may be more informative and thus we introduce a non-negative hyper-parameter $\alpha$ to sample a link with a probability proportional to $\exp(\alpha(t - t_\mathrm{p}))$ where $t, t_\mathrm{p}$ are the timestamps of this link and the link previously sampled respectively. A large $\alpha$ can emphasize more on recent links while zero $\alpha$ leads to uniform sampling. In Sec.4.4, we will discuss an efficient sampling strategy for the step 5 in Alg.1, which avoids computing those probabilities by visiting the entire historical links.

Then, given a link $\{u_0, v_0\}$ and a time $t_0$, we use Alg. 1 to collect $M$ many $m$-step walks starting from both $u_0$ and $v_0$, and record them in $S_u$ and $S_v$ respectively. For convenience, a walk $W$ from a starting node $w_0 \in \{u, v\}$ can be represented as Eq.1.

**Set-based Anonymization**. Based on $S_u$ and $S_v$, we may anonymize each node identity $w$ that appears on at least one walk in $S_u \cup S_v$ and design relative node identity $I_{CAW}(w; \{S_u, S_v\})$ for $w$. Our design has the following consideration. $I_{AW}$ (Eq.2) only depends on a single path, which results from the original assumption that any two AWs do not even share the name space (*i.e.*, node identities) (Micali & Zhu, 2016). However, in our case, node identities are actually accessible, though an inductive model is not allowed to use them directly. Instead, correlation across different walks could be a key to reflect laws of network dynamics: Consider the case when the link $\{u, v\}$ happens only if there is another node appearing in multiple links connected to $u$ (Fig. 4). Therefore, we propose to use node identities to first establish such correlation and then remove the original identities.

Specifically, we define $I_{CAW}(w; \{S_u, S_v\})$ as follows: For $w_0 \in \{u, v\}$, let $g(w, S_{w_0}) \in \mathbb{Z}^{m+1}$ count the times in $S_{w_0}$ node $w$ appears at certain positions, i.e., $g(w, S_{w_0})[i] \triangleq |\{W | W \in S_{w_0}, w = W[i][0]\}|$ for $i \in \{0, 1, ..., m\}$. Further, define

$$I_{CAW}(w; \{S_u, S_v\}) \triangleq \{g(w, S_u), \ g(w, S_v)\}. \tag{3}$$

Essentially, $g(w, S_u)$ and $g(w, S_v)$ encode the correlation between walks within $S_u$ and $S_v$ respectively and the set operation in Eq.3 establishes the correlation across $S_u$ and $S_v$. For the case in Fig.3, suppose $S_a, S_b, S_{a'}, S_{b'}$ include all historical one-step walks (before $t_3$) starting from $a, b, a', b'$ respectively. Then, it is easy to show that $I_{CAW}(a; \{S_a, S_b\}) \neq I_{CAW}(a'; \{S_{a'}, S_b\})$ that allows differentiating $a$ and $a'$, while TGAT (Xu et al., 2020) as discussed in Sec.2 fails. From the network-motif point of view, $I_{CAW}$ not only encodes each network motif that corresponds to one single walk as $I_{AW}$ does but also establish the correlation among these network motifs. $I_{AW}$ cannot establish the correlation between motifs and will also fail to distinguish $a$ and $a'$ in Fig.3. We see it as a significant breakthrough as such correlation often gets neglected in previous works that directly count motifs or adopt AW-type anonymization $I_{AW}$.

Later, we use $I_{CAW}(w)$ for simplicity when the reference set $\{S_u, S_v\}$ can be inferred from the context. Then, each walk $W$ (Eq.1) can be anonymized as

$$\hat{W} = ((I_{CAW}(w_0), t_0), (I_{CAW}(w_1), t_1), ..., (I_{CAW}(w_m), t_m)). \tag{4}$$

The following theorem indicates that $I_{CAW}$ does not depend on node identities to guarantee the inductive property of the models, which can be easily justified.

**Theorem 4.1.** For two pairs of walk sets $\{S_u, S_v\}$ and $\{S_{u'}, S_{v'}\}$, if there exists a bijective mapping $\pi$ between node identities such that each walk $W$ in $S_u \cup S_v$ can be bijectively mapped to one walk $W'$ in $S_{u'} \cup S_{v'}$ according to $\pi(W[i][0]) = W'[i][0]$ for all $i \in [0, m]$. Then $I_{CAW}(w|\{S_u, S_v\}) = I_{CAW}(\pi(w)|\{S_{u'}, S_{v'}\})$ for all nodes $w$ that appear in at least one walk in $S_u \cup S_v$.

## 4.3 NEURAL ENCODING FOR CAUSAL ANONYMOUS WALKS

After we collect CAWs, neural networks can be conveniently leveraged to extract their structural (in $I_{CAW}(\cdot)$) and fine-grained temporal information by encoding CAWs: We will propose the model CAW-N to first encode each walk $\hat{W}$ (Eq.4) and then aggregate all encoded walks in $S_u \cup S_v$.

| Measurement | Reddit | Wikipedia | MOOC | Social Evo. | Enron | UCI |
|---|---|---|---|---|---|---|
| nodes & temporal links | 10,985 & 672,447 | 9,227 & 157,474 | 7145 & 411,749 | 184 & 125,235 | 74 & 2,099,520 | 1,899 & 59,835 |
| attributes for nodes & links | 172 & 172 | 172 & 172 | 0 & 4 | 0 & 0 | 0 & 0 | 0 & 0 |
| avg. link stream intensity $\tau$ | $4.57 \times 10^{-5}$ | $1.27 \times 10^{-5}$ | $4.48 \times 10^{-5}$ | $6.50 \times 10^{-5}$ | $4.98 \times 10^{-3}$ | $3.59 \times 10^{-5}$ |

Table 1: Summary of dataset statistics. Average link stream intensity is calculated by $2|E|/(|V|T)$, where $T$ is the total time range of all edges in unit of seconds, $|V|$ and $|E|$ are number of nodes and temporal links.

**Encode $\hat{W}$.** Note that each walk is a sequence of node-time pairs. If we encode each node-time pair and plug those pairs in a sequence encoder, *e.g.*, RNNs, we obtain the encoding of $\hat{W}$:

$$\text{enc}(\hat{W}) = \text{RNN}(\{f_1(I_{CAW}(w_i)) \oplus f_2(t_{i-1} - t_i)\}_{i=0,1,...,m}), \text{ where } t_{-1} = t_0, \tag{5}$$

where $f_1, f_2$ are two encoding function on $I_{CAW}(w_i)$ and $t_{i-1} - t_i$ respectively. One may use transformer networks instead of RNNs to encode the sequences but as the sequences in our case are not long $(1 \sim 5)$, RNNs have achieved good enough performance. Now, we specify the two encoding functions $f_1(I_{CAW}(w_i))$ and $f_2(t_{i-1} - t_i)$ as follows. Recall the definition of $I_{CAW}(w_i)$ (Eq.3).

$$f_1(I_{CAW}(w_i)) = \text{MLP}(g(w, S_u)) + \text{MLP}(g(w, S_v)), \quad \text{where two MLPs share parameters.} \tag{6}$$

Here the encoding of $I_{CAW}(w_i)$ adopts the sum-pooling as the order of $u, v$ is not relevant. For $f_2(t)$, we adopt random Fourier features to encode time (Xu et al., 2019; Kazemi et al., 2019) which may approach any positive definite kernels according to the Bochner's theorem (Bochner, 1992).

$$f_2(t) = [\cos(\omega_1 t), \sin(\omega_1 t), ..., \cos(\omega_d t), \sin(\omega_d t)], \text{ where } \omega_i\text{'s are learnable parameters.} \tag{7}$$

**Encode $S_u \cup S_v$.** After encoding each walk in $S_u \cup S_v$, we aggregate all these walks to obtain the final representation $\text{enc}(S_u \cup S_v)$ for prediction. We suggest to use either mean-pooling for algorithmic efficiency or self-attention (Vaswani et al., 2017) followed by mean-pooling to further capture subtle interactions between different walks. Specifically, suppose $\{\hat{W}_i\}_{1 \leq i \leq 2M}$ are the $2M$ CAWs in $S_u \cup S_v$ and each $\text{enc}(\hat{W}_i) \in \mathbb{R}^{d \times 1}$. We set $\text{enc}(S_u \cup S_v)$ as

- Mean-AGG($S_u \cup S_v$): $\frac{1}{2M} \sum_{i=1}^{2M} \text{enc}(\hat{W}_i)$.
- Self-Att-AGG($S_u \cup S_v$): $\frac{1}{2M} \sum_{i=1}^{2M} \text{softmax}(\{\text{enc}(\hat{W}_i)^T Q_1 \text{enc}(\hat{W}_j)\}_{1 \leq j \leq n}) \text{enc}(\hat{W}_i) Q_2$ where $Q_1, Q_2 \in \mathbb{R}^{d \times d}$ are two learnable parameter matrices.

We add 2-layer perceptron over $\text{enc}(S_u \cup S_v)$ to make the final link prediction.

### 4.4 EXTENSION AND DISCUSSION

**Attributed nodes/links and directed links**. In some real networks, nodes or links may have attributes available, e.g., the message content in the case of SMS networks. In this case, the walk in Eq.1 can be associated with node/link attributes $X_0, X_1, ..., X_m$ where $X_i$ refers to the attributes on link $(\{w_{i-1}, w_i\}, t_i)$ or on the node $w_i$ or a concatenation of these two parts. Note that the direction of a link can also be viewed as a binary link attribute, where a 2-dimensional one-hot encoding can be used. To incorporate such information, we only need to change $\text{enc}(\hat{W})$ (Eq.5) as

$$\text{enc}(\hat{W}) = \text{RNN}(\{f_1(I_{CAW}(w_i)) \oplus f_2(t_{i-1} - t_i) \oplus X_i\}_{i=0,1,...,m}), \text{ where } t_{-1} = t_0. \tag{8}$$

Since $f_1(I_{CAW}(w_i))$ is a strong signal, in practice it is optional to use another RNN to encode its own dynamics. The derived encoding is then concatenated with $\text{enc}(\hat{W})$ to obtain the enhanced final encoding of $\hat{W}$.

**Efficient link sampling**. A naive implementation of the link sampling in step 5 in Alg.1 is to compute and normalize the sampling probabilities of all links in $\mathcal{E}_{w_p, t_p}$, which requires to memorize all historical links and costs much time and memory. To solve this problem, we propose a sampling strategy (Appendix A) with expected time and memory complexity $\min\{\frac{2\tau}{\alpha} + 1, |E_{w_p, t_p}|\}$ if links in $\mathcal{E}_{w_p, t_p}$ come in by following a Poisson process with intensity $\tau$ (Last & Penrose, 2017). This means that our sampling strategy with a positive $\alpha$ allows the model only recording $O(\frac{\tau}{\alpha})$ recent links for each node instead of the entire history. Our experiments in Sec.5.3 show that $\alpha$ that achieves the best prediction performance makes $\frac{\tau}{\alpha} \approx 5$ in different datasets. Since the time and memory complexity do not increase with respect to the number of links, our model can be used for online training and inference. Note that the $\alpha = 0$ case reduces to uniform sampling, mostly adopted by previous methods (Xu et al., 2020), which requires to record the entire history and thus is not scalable.

## 5 EXPERIMENTS

### 5.1 EXPERIMENTAL SETUP

**CAW-N Variants**. We test CAW-N-mean and CAW-N-attn which uses mean and attention pooling respectively to encode $S_u \cup S_v$ (Sec. 4.3). Their code is provided in the supplement.

**Baselines**. Our method is compared with six previous state-of-the-art baselines on representation learning of temporal networks. They can be grouped into two categories based on their input data structure: (1) Snapshot-based methods, including DynAERNN (Goyal et al., 2020), VGRNN (Haji-ramezanali et al., 2019) and EvolveGCN (Pareja et al., 2020); (2) Stream-based methods, including TGAT (Xu et al., 2020), JODIE (Kumar et al., 2019) and DyRep (Trivedi et al., 2019). We give their detailed introduction in Appendix C.2.2. For the snapshot-based methods, we view the link aggregation as a way to preprocess historical links. We adopt the aggregation ways suggested in their papers. These models are trained and evaluated over the same link sets as the stream-based methods.

**Dataset**. We use six real-world public datasets: Wikipedia is a network between wiki pages and human editors. Reddit is a network between posts and users on subreddits. MOOC is a network of students and online course content units. Social Evolution is a network recording the physical proximity between students. Enron is a email communication network. UCI is a network between online posts made by students. We summarize their statistics in Tab.1 and give their detailed description and access in Appendix C.1.

**Evaluation Tasks**. Two types of tasks are for evaluation: transductive and inductive link prediction.

*Transductive link prediction task* allows temporal links between all nodes to be observed up to a time point during the training phase, and uses all the remaining links after that time point for testing. In our implementation, we split the total time range $[0, T]$ into three intervals: $[0, T_{train})$, $[T_{train}, T_{val})$, $[T_{val}, T]$. links occurring within each interval are dedicated to training, validation, and testing set, respectively. For all datasets, we fix $T_{train}/T$=0.7, and $T_{val}/T$=0.85 .

*Inductive link prediction task* predicts links associated with nodes that are not observed in the training set. There are two types of such links: 1) **"old vs. new"** links, which are links between an observed node and an unobserved node; 2) **"new vs. new"** links, which are links between two unobserved nodes. Since these two types of links suggest different types of inductiveness, we distinguish them by reporting their performance metrics *separately*. In practice, we follow two steps to split the data: 1) we use the same setting of the transductive task to first split the links chronologically into training / validation / testing sets; 2) we randomly select 10% nodes, remove any links associated with them from the training set, and remove any links *not* associated with them in the validation and testing sets.

Following most baselines, we randomly sample an equal amount of negative links and consider link prediction as a binary classification problem. For fair comparison, we use the same evaluation procedures for all baselines, including the snapshot-based methods.

**Training configuration**. We use binary cross entropy loss and Adam optimizer to train all the models, and early stopping strategy to select the best epoch to stop training. For hyperparameters, we primarily tune those that control the CAW sampling scheme including the number $M$, the length $m$ of CAWs and the time decay $\alpha$. We will investigate their sensitivity in Sec.5.3. For all baselines, we adapt their implemented models into our evaluation pipeline and extensively tune them. Detailed description of all models' tuning can be found in Appendix C. Finally, we adopt two metrics to evaluate the models' performance: Area Under the ROC curve (AUC) and Average Precision (AP).

### 5.2 RESULTS AND DISCUSSION

We report AUC scores in Tab.2, and report AP scores in Tab.6 of Appendix D.1. In the inductive setting and especially with "new vs new" links, our models significantly outperform all baselines on all datasets. On average, our best method improves over the strongest baseline by 14.46% (new vs. new) and 3.49% (old vs. new) in relative, or reduces the error (= 1 - AUC) by 69.73% (new vs. new) and 58.63% (old vs. new). Noticeably, out method achieves almost perfect AUCs on UCI's "new vs. new" edges, when all baselines' performance is below 0.8.

Even in transductive setting where the baselines claim their primary contribution, our approaches still significantly outperform them on five out of six datasets. Note that our models achieve almost perfect scores on Reddit and Wikpedia when the baselines are far from perfect. Meanwhile, the strongest baseline on these two attributed datasets, TGAT (Xu et al., 2020), suffers a lot on all the other datasets where informative node / link attributes become unavailable.

| Task | | Methods | Reddit | Wikipedia | MOOC | Social Evo. | Enron | UCI |
|---|---|---|---|---|---|---|---|---|
| Inductive | new v.s. new | DynAERNN | $57.51 \pm 2.54$ | $55.16 \pm 1.15$ | $60.85 \pm 1.61$ | $52.00 \pm 0.16$ | $51.57 \pm 2.63$ | $50.20 \pm 2.78$ |
| | | JODIE | $72.49 \pm 0.38$ | $70.78 \pm 0.75$ | $80.04 \pm 0.28^\dagger$ | $87.66 \pm 0.12^\dagger$ | $73.99 \pm 2.54^\dagger$ | $64.77 \pm 0.75$ |
| | | DyRep | $62.37 \pm 1.49$ | $67.07 \pm 1.26$ | $74.07 \pm 1.88$ | $83.92 \pm 0.02$ | $69.74 \pm 0.44$ | $63.76 \pm 4.67$ |
| | | VGRNN | $61.93 \pm 0.72$ | $60.64 \pm 0.68$ | $63.01 \pm 0.29$ | $66.30 \pm 0.84$ | $61.35 \pm 1.10$ | $61.35 \pm 1.10$ |
| | | EvolveGCN | $63.31 \pm 0.53$ | $58.01 \pm 0.16$ | $52.31 \pm 4.14$ | $46.95 \pm 0.85$ | $42.53 \pm 2.12$ | $76.65 \pm 0.63^\dagger$ |
| | | TGAT | $94.96 \pm 0.88^\dagger$ | $93.53 \pm 0.84^\dagger$ | $70.10 \pm 0.35$ | $53.27 \pm 1.16$ | $63.34 \pm 2.95$ | $76.36 \pm 1.48$ |
| | | **CAW-N-mean** | $\mathbf{98.30 \pm 0.71^*}$ | $\mathbf{96.36 \pm 0.48^*}$ | $\mathbf{90.29 \pm 0.82^*}$ | $\mathbf{93.81 \pm 0.69^*}$ | $\mathbf{94.26 \pm 0.62^*}$ | $\mathbf{99.62 \pm 0.34^*}$ |
| | | **CAW-N-attn** | $\mathbf{98.11 \pm 0.58^*}$ | $\mathbf{97.83 \pm 0.67^*}$ | $\mathbf{90.40 \pm 0.75^*}$ | $\mathbf{94.55 \pm 0.81^*}$ | $\mathbf{93.53 \pm 0.63^*}$ | $\mathbf{100.00 \pm 0.00^*}$ |
| | new v.s. old | DynAERNN | $58.79 \pm 3.01$ | $57.97 \pm 2.38$ | $80.99 \pm 1.35$ | $52.31 \pm 0.59$ | $54.36 \pm 1.48$ | $52.26 \pm 1.36$ |
| | | JODIE | $76.33 \pm 0.03$ | $74.65 \pm 0.06$ | $87.40 \pm 1.71$ | $91.80 \pm 0.01^\dagger$ | $85.24 \pm 0.08$ | $69.95 \pm 0.11$ |
| | | DyRep | $66.13 \pm 1.07$ | $76.72 \pm 0.19$ | $88.23 \pm 1.20^\dagger$ | $87.98 \pm 0.45$ | $94.39 \pm 0.32^\dagger$ | $93.28 \pm 0.96^\dagger$ |
| | | VGRNN | $54.11 \pm 0.74$ | $62.93 \pm 0.69$ | $60.10 \pm 0.88$ | $64.66 \pm 0.41$ | $68.71 \pm 0.92$ | $62.39 \pm 1.08$ |
| | | EvolveGCN | $65.61 \pm 0.37$ | $56.29 \pm 2.17$ | $50.20 \pm 1.92$ | $50.73 \pm 1.36$ | $42.53 \pm 2.13$ | $70.78 \pm 0.22$ |
| | | TGAT | $97.25 \pm 0.18^\dagger$ | $95.47 \pm 0.17^\dagger$ | $69.30 \pm 0.08$ | $54.22 \pm 1.28$ | $58.76 \pm 1.18$ | $74.19 \pm 0.88$ |
| | | **CAW-N-mean** | $\mathbf{99.88 \pm 0.04^*}$ | $\mathbf{98.94 \pm 0.05^*}$ | $\mathbf{90.88 \pm 0.54^*}$ | $\mathbf{95.15 \pm 0.40^*}$ | $\mathbf{94.76 \pm 1.05^*}$ | $\mathbf{99.04 \pm 0.34^*}$ |
| | | **CAW-N-attn** | $\mathbf{99.93 \pm 0.03^*}$ | $\mathbf{99.61 \pm 0.25^*}$ | $\mathbf{90.89 \pm 0.56^*}$ | $\mathbf{95.74 \pm 0.68^*}$ | $93.43 \pm 1.41$ | $\mathbf{98.99 \pm 0.44^*}$ |
| Transductive | | DynAERNN | $83.37 \pm 1.48$ | $71.00 \pm 1.10$ | $89.34 \pm 0.24$ | $67.78 \pm 0.80$ | $63.11 \pm 1.13$ | $83.72 \pm 1.79$ |
| | | JODIE | $87.71 \pm 0.02$ | $88.43 \pm 0.02$ | $90.50 \pm 0.01^\dagger$ | $89.78 \pm 0.04$ | $89.36 \pm 0.06$ | $74.63 \pm 0.11$ |
| | | DyRep | $67.36 \pm 1.23$ | $77.40 \pm 0.13$ | $90.49 \pm 0.03$ | $90.85 \pm 0.01^\dagger$ | $96.71 \pm 0.04^\dagger$ | $95.23 \pm 0.25^\dagger$ |
| | | VGRNN | $51.89 \pm 0.92$ | $71.20 \pm 0.65$ | $90.03 \pm 0.32$ | $78.28 \pm 0.69$ | $93.84 \pm 0.58$ | $89.43 \pm 0.27$ |
| | | EvolveGCN | $58.42 \pm 0.52$ | $60.48 \pm 0.47$ | $50.36 \pm 0.85$ | $60.36 \pm 0.65$ | $74.02 \pm 0.31$ | $78.30 \pm 0.22$ |
| | | TGAT | $96.65 \pm 0.06^\dagger$ | $96.36 \pm 0.05^\dagger$ | $72.09 \pm 0.29$ | $56.63 \pm 0.55$ | $60.88 \pm 0.37$ | $77.67 \pm 0.27$ |
| | | **CAW-N-mean** | $\mathbf{99.97 \pm 0.01^*}$ | $\mathbf{99.91 \pm 0.04^*}$ | $\mathbf{91.99 \pm 0.72^*}$ | $\mathbf{94.12 \pm 0.15^*}$ | $93.53 \pm 0.73$ | $\mathbf{95.90 \pm 0.71}$ |
| | | **CAW-N-attn** | $\mathbf{99.98 \pm 0.01^*}$ | $\mathbf{99.89 \pm 0.03^*}$ | $\mathbf{92.38 \pm 0.58^*}$ | $\mathbf{94.79 \pm 0.16^*}$ | $95.93 \pm 0.39$ | $\mathbf{98.45 \pm 0.49^*}$ |

Table 2: Performance in AUC (mean in percentage $\pm 95\%$ confidence level.) $\dagger$ highlights the best baselines. $^*$, **bold font**, **bold font**$^*$ respectively highlights the case where our models' performance exceeds the best baseline on average, by $70\%$ confidence, by $95\%$ confidence.

| No. | Ablation | Wikipedia | UCI | Social Evo. |
|---|---|---|---|---|
| 1. | original method (CAW-N-mean) | $98.49 \pm 0.38$ | $99.12 \pm 0.33$ | $94.54 \pm 0.69$ |
| 2. | remove $f_1(I_{CAW})$ | $96.28 \pm 0.66$ | $79.45 \pm 0.71$ | $53.69 \pm 0.21$ |
| 3. | remove $f_2(t)$ | $97.99 \pm 0.13$ | $95.00 \pm 0.42$ | $71.93 \pm 2.32$ |
| 4. | remove $f_1(I_{CAW})$, $f_2(t)$ | $88.94 \pm 0.85$ | $50.01 \pm 0.02$ | $50.00 \pm 0.00$ |
| 5. | replace $f_1(I_{CAW})$ by one-hot$(I_{AW})$ | $96.55 \pm 0.21$ | $85.59 \pm 0.34$ | $68.47 \pm 0.86$ |
| 6. | fix $\alpha = 0$ | $75.10 \pm 3.12$ | $87.13 \pm 0.49$ | $78.01 \pm 0.69$ |

Table 3: Ablation study with CAW-N-mean. AUC scores on *all inductive* test links are reported.

Comparing the two training settings, we observe that the performance of four baselines (JODIE, VGRNN, DynAERNN,DyRep) drop significantly when transiting from the transductive setting to the inductive one, as they mostly record node identities either explicitly (JODIE, VGRNN, DynAERNN) or implicitly (DyRep). TGAT and EvolveGCN do not use node identities and thus their performance gaps between the two settings are small, while they sometimes do not perform well in the transductive setting, as they encounter the ambiguity issue in Fig. 3. In contrast, our methods perform well in both the transductive and inductive settings. We attribute this superiority to the anonymization procedure: the set-based relative node identities well capture the correlation between walks to make good prediction while removing the original node identities to keep entirely inductive. Even when the network structures greatly change and new nodes come in as long as the network evolves according to the same law as the network used for training, CAW-N will always work. Comparing CAW-N-mean and CAW-N-attn, we see that our attention-based variant outperforms the mean-pooling variant, albeit at the cost of high computation complexity. Also note that the strongest baselines on all datasets are stream-based methods, which indicates that the aggregation of links into network snapshots may remove some useful time information (see more discussion in Appendix D.2).

We further conduct ablation studies on Wikipedia (attributed), UCI (non-attributed), and Social Evolution (non-attributed), to validate effectiveness of critical components of our model. Tab. 3 shows the results. By comparing Ab.1 with Ab.2, 3 and 4 respectively, we observe that our proposed node anonymization and encoding, $f_1(I_{CAW})$, contributes most to the performance, though the time encoding $f_2(t)$ also helps. Comparing performance across different datasets, we see that the impact of ablation is more prominent when informative node/link attributes are unavailable such as with UCI and Social Evolution. Therefore, in such scenarios our CAWs are highly crucial. In Ab.5, we replace our proposed $I_{CAW}$ with $I_{AW}$ (Eq.2), which is used in standard AWs, and we use one-hot encoding of node new identities $I_{AW}$. We see by comparing Ab.1, 2, and 5 that such anonymization process is significantly less effective than our $I_{CAW}$, though it helps to some extent. Finally, Ab.6 suggests that entirely uniform sampling of the history may hurt performance.

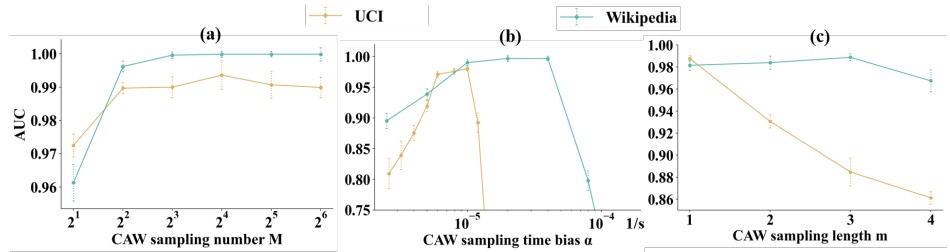

Figure 5: Hyperparameter sensitivity in CAW sampling. AUC on *all inductive* test links are reported.

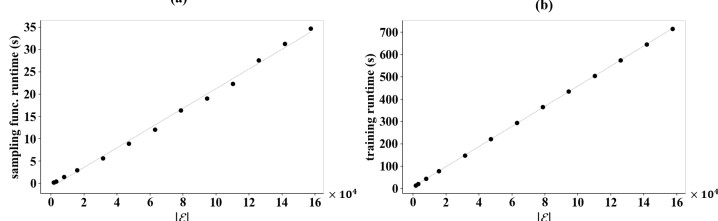

Figure 6: Complexity evaluation: The accumulated runtime of (a) temporal random walk extraction (Alg.1) and (b) the entire CAW-N training, timed over one epoch on Wikipedia (using different $|\mathcal{E}|$ for training).

### 5.3 HYPERPARAMETER INVESTIGATION OF CAW SAMPLING

We systematically analyze the effect of hyperparameters used in CAW sampling schemes, including sampling number $M$, temporal decay coefficient $\alpha$ and walk length $m$. The experiments are conducted on UCI and Wikipedia datasets using CAW-N-mean. When investigating each hyperparameter, we set the rest two to an optimal value found by grid search, and report the mean AUC performance on all inductive test links (*i.e.* old vs. new, new vs. new) and their 95% confidence intervals.

The results are summarized in Fig.5. From (a), we observe that only a small number of sampled CAWs are needed to achieve a competitive performance. Meanwhile, the performance gain is saturated as the sampling number increases. We analyze the temporal decay $\alpha$ in (b): $\alpha$ usually has an optimal interval, whose values and length also vary with different datasets to capture the different levels of temporal dynamics; a small $\alpha$ suggests an almost uniform sampling of interaction history, which hurts the performance; an overly large $\alpha$ also damages the model, since it makes the model only sample the most recent few interactions for computation and blind to the rest. Based on our efficient sampling strategy (Sec.4.4), we may combine the optimal $\alpha$ with the average link intensity $\tau$ (Tab.1), and concludes that CAW-N only needs to online record and sample from about a constant times about 5 ($\approx \frac{\tau}{\alpha}$) most recent links for each node. Plot (c) suggests that the performance may peak at a certain CAW length, while the exact value may vary with datasets. Longer CAWs indicate that the corresponding networks evolve according to more complicated laws encoded in higher-order motifs.

### 5.4 COMPLEXITY EVALUATION

We examine how the runtime of CAW-N depends on the number of edges $|\mathcal{E}|$ used for training. We record the runtimes of CAW-N for training one epoch on the Wikipedia datasets using $M = 32, m = 2$ with batch-size=32. Specifics of the computing infrastructure are given in Appendix C.5. Fig. 6 (a) shows the accumulated runtime of executing the random walk extraction *i.e.* Alg.1 only. It well aligns with our theoretical analysis (Thm. A.2) that each step of the random walk extraction has constant complexity (*i.e.* accumulated runtime linear with $|\mathcal{E}|$). Plot (b) shows the entire runtime for one-epoch training, which is also linear with $|\mathcal{E}|$. Note that $O(|\mathcal{E}|)$ is the time complexity that one at least needs to pay. The study demonstrates our method is scalable to long edge streams.

## 6 CONCLUSION

We proposed CAW-N to inductively represent the dynamics of temporal networks. CAW-N uses CAWs to implicitly extract network motifs via temporal random walks and adopts novel set-based anonymization to establish the correlation between network motifs. The success of CAW-N points out many promising future research directions on temporal networks: Pairing CAW-N with neural network interpretation techniques (Montavon et al., 2018) may give a chance to automatically discover larger and meaningful motifs/patterns of temporal networks; CAW-N may also be generalized to predict high-order structures (*e.g.*, triangles) that correspond to some function units of temporal networks from different domains (Benson et al., 2016; 2018; Zitnik et al., 2019).

ACKNOWLEDGMENTS

We thank Jiaxuan You and Rex Ying for their helpful discussion on the idea of Causal Anonymous Walks. We also thank Rok Sosič, Camilo Andres Ruiz and Maria Brbić for providing insightful feedback on the abstract. We also gratefully acknowledge the support of DARPA under Nos. FA865018C7880 (ASED), N660011924033 (MCS); ARO under Nos. W911NF-16-1-0342 (MURI), W911NF-16-1-0171 (DURIP); NSF under Nos. OAC-1835598 (CINES), OAC-1934578 (HDR), CCF-1918940 (Expeditions), IIS-2030477 (RAPID); Stanford Data Science Initiative, Wu Tsai Neurosciences Institute, Chan Zuckerberg Biohub, Amazon, Boeing, JPMorgan Chase, Docomo, Hitachi, JD.com, KDDI, NVIDIA, Dell. J. L. is a Chan Zuckerberg Biohub investigator.

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

## A  EFFICIENT LINK SAMPLING

Our efficient link sampling strategy contains two subroutines – Online probability computation (Alg.2) and Iterative sampling (Alg.3). The Online probability computation subroutine Alg.2 essentially works online to assign each new incoming link $(\{u, v\}, t)$ with a pair of probabilities $\{p_{u,t}, p_{v,t}\}$ such that

$$p_{u,t} = \frac{\exp(\alpha t)}{\sum_{(e,t') \in E_{u,t}} \exp(\alpha t')}, \quad p_{v,t} = \frac{\exp(\alpha t)}{\sum_{(e,t') \in E_{v,t}} \exp(\alpha t')}. \tag{9}$$

These probabilities will be used later in sampling (Alg.3) and do not need to be updated any more.

---

**Algorithm 2:** Online probability computation $(G, \alpha)$

---

1   Initialize $V \leftarrow \emptyset, \Omega \leftarrow \emptyset$;
2   **for** $(\{u, v\}, t) \in \mathcal{E}$ **do**
3      **for** $w \in \{u, v\}$ **do**
4         **if** $w \notin V$ **then**
5             $V \leftarrow V \cup \{w\}$;
6             $P_w \leftarrow \exp(\alpha t)$;
7         **else**
8             Find $P_w \in \Omega$;
9             $P_w \leftarrow P_w + \exp(\alpha t)$;
10         $p_{w,t} \leftarrow \frac{\exp(\alpha t)}{P_w}$;
11         $\Omega \leftarrow \Omega \cup \{P_w\}$;
12      Assign two probability scores: $(\{(u, p_{u,t}), (v, p_{v,t})\}, t,) \leftarrow (\{u, v\}, t)$ ;

---

The Iterative sampling subroutine Alg.3 is an efficient implementation of step 5 in Alg.1. We may first show that the sampling probability of a link $(e, t)$ in $E_{w_p, t_p}$ is proportional to $\exp(\alpha(t - t_p))$ in Prop.A.1.

---

**Algorithm 3:** Iterative Sampling $(\mathcal{E}, \alpha, w_p, t_p)$

---

1   Initialize $V \leftarrow \emptyset, \Omega \leftarrow \emptyset$;
2   **for** $(e, t) \in \mathcal{E}_{w_p, t_p}$ *with an decreasing order of* $t$ **do**
3      Sample $a \sim \text{Unif}[0, 1]$;
4      $p_{w_p, t}$ is the score of this link related to $w_p$ obtained from Alg.2;
5      **if** $a < p_{w_p, t}$ **then**
6         Return $(e, t)$;
7   Return $(\{w_p, X\}, t_X)$;

---

**Proposition A.1.** Based on the probabilities (Eq.9) pre-computed by Alg.2, Alg.3 will sample a link $(e, t)$ in $\mathcal{E}_{w_p, t_p}$ with probability proportional to $\exp(\alpha(t - t_p))$.

*Proof.* To show this, we first order the timestamps of links in $\mathcal{E}_{w_p, t_p}$ between $[t, t_p]$ as $t = t'_0 < t'_1 < t'_2 < \cdots < t'_k < t_p$ where there exists an link $(e', t'_i) \in \mathcal{E}_{w_p, t_p}$ for $t'_i$. Then, the probability to sample a link $(e, t) \in \mathcal{E}_{w_p, t_p}$ satisfies

$$p = p_{w_p, t} \times \prod_{i=1}^{k} \left(1 - p_{w_p, t'_i}\right)$$

$$= \frac{\exp(\alpha t)}{\sum_{(e',t') \in \mathcal{E}_{w_p, t}} \exp(\alpha t')} \times \prod_{i=1}^{k} \frac{\sum_{(e',t') \in \mathcal{E}_{w_p, t'_{i-1}}} \exp(\alpha t')}{\sum_{(e',t') \in \mathcal{E}_{w_p, t'_i}} \exp(\alpha t')}$$

$$= \frac{\exp(\alpha t)}{\sum_{(e',t') \in \mathcal{E}_{w_p, t_p}} \exp(\alpha t')} = \frac{\exp(\alpha(t - t_p))}{\sum_{(e',t') \in \mathcal{E}_{w_p, t_p}} \exp(\alpha(t' - t_p))},$$

which is exactly the probability that we need. $\qquad\square$

We have the following Thm.A.2 with a weak assumption that evaluates the complexity of Alg.3. We assume that links come in by following a Poisson point process with intensity $\tau$, which is a frequently used assumption to model communication networks (Lavenberg, 1983). This result indicates that if $\alpha > 0$, for each node, we only need to record the most recent $O(\frac{\tau}{\alpha})$ links to sample. This result is important as it means our method can do online training and inference with time and memory complexity that are not related to the total number of links.

**Theorem A.2.** If the links that are connected to $w_\mathrm{p}$ appear by following a Poisson point process with intensity $\tau$. Then, the expected number of iterations of Alg.3 is bounded by $\min\{\frac{2\tau}{\alpha} + 1, |\mathcal{E}_{w_\mathrm{p},t_\mathrm{p}}|\}$.

*Proof.* The number of iterations of Alg.3 is always bounded by $|\mathcal{E}_{w_\mathrm{p},t_\mathrm{p}}|$. So we only need to prove that the expected number of interactions of Alg.3 is bounded by $\frac{2\tau}{\alpha} + 1$.

To show this, we order the timestamps of links in $\mathcal{E}_{w_\mathrm{p},t_\mathrm{p}}$ between $[0, t_\mathrm{p}]$ as $0 = t_0 < t_1 < t_2 < \cdots < t_k < t_\mathrm{p}$ where there exists an link $(e, t_i) \in \mathcal{E}_{w_\mathrm{p},t_\mathrm{p}}$ for $t_i$. Further define we define $Z_i = \exp(\alpha(t_i - t_{i-1}))$ for $i \in [1, k]$.

Due to the definition of Poisson process, we know that each time difference in $\{t_i - t_{i-1}\}_{1 \leq i \leq k}$ follows i.i.d. exponential distribution with parameter $\tau$. Therefore, $\{Z_i\}_{1 \leq i \leq k}$ are also i.i.d.. Let $\psi = \mathbb{E}(Z_i^{-1}) = \frac{\tau}{\alpha + \tau}$.

The probability that Alg.3 runs $j$ iterations is equal to the probability that the link with timestamp $t_{k+1-j}$) gets sampled. That is

$$\mathbb{P}(\mathrm{iter} = j) = \frac{\prod_{i=1}^{k+1-j} Z_i}{\sum_{h=1}^{k} \prod_{i=1}^{k+1-h} Z_i}.$$

Therefore, the expected number of iterations is

$$\mathbb{E}(\mathrm{iter}) = \mathbb{E}\left[\frac{\sum_{j=1}^{k} j \prod_{i=1}^{k+1-j} Z_i}{\sum_{h=1}^{k} \prod_{i=1}^{k+1-h} Z_i}\right] = \sum_{j=1}^{k} j\mathbb{E}\left[\frac{\prod_{i=1}^{j-1} Z_i'}{\sum_{h=1}^{k} \prod_{i=1}^{h-1} Z_i'}\right]. \tag{10}$$

where $Z_i' = Z_{k+1-i}^{-1}$ and $\prod_{i=1}^{0} Z_i' = 1$. Next we will prove that each item in right-hand-side of Eq.10 satisfies

$$j\mathbb{E}\left[\frac{\prod_{i=1}^{j-1} Z_i'}{\sum_{h=1}^{k} \prod_{i=1}^{h-1} Z_i'}\right] \leq [1 + (j-1)(1-\psi)]\psi^{j-1}. \tag{11}$$

If this is true, then

$$\mathbb{E}(\mathrm{iter}) \leq \sum_{j=1}^{k}[1 + (j-1)(1-\psi)]\psi^{j-1} = \sum_{j=1}^{k} \psi^{j-1} + \sum_{j=1}^{k}(j-1)(1-\psi)\psi^{j-1}$$

$$\leq \frac{1}{1-\psi} + \frac{\psi}{1-\psi} = \frac{2\tau}{\alpha} + 1.$$

Now, let us prove Eq.11. For $j = 1$, Eq.11 is trivial. For $j > 1$,

$$\mathbb{E}\left[\frac{\prod_{i=1}^{j-1} Z_i'}{\sum_{h=1}^{k} \prod_{i=1}^{h-1} Z_i'}\right] \leq \mathbb{E}\left[\frac{\prod_{i=1}^{j-1} Z_i'}{1 + \sum_{h=2}^{j} \prod_{i=1}^{h-1} Z_i'}\right] \leq \frac{\prod_{i=1}^{j-1} \mathbb{E}(Z_i')}{1 + \sum_{h=2}^{j} \prod_{i=1}^{h-1} \mathbb{E}(Z_i')} = \frac{\psi^{j-1}}{\sum_{i=0}^{j-1} \psi^i}, \tag{12}$$

where the second inequality is due to the Jensen's inequality and the fact that for any positive $c_1, c_2$, $\frac{x}{c_1 + c_2 x}$ is concave with respect to $x$. Moreover, we also have

$$[1 + (j-1)(1-\psi)]\sum_{i=0}^{j-1} \psi^i = j + \sum_{i=1}^{j-1} \psi^i - (j-1)\psi^j \geq j. \tag{13}$$

Combining Eq.12 and Eq.13, we prove Eq.11, which concludes the proof. $\qquad\square$

## B    Tree-structured sampling

We may further decrease the sampling complexity by revising Alg. 1 into tree-structured sampling. Alg. 1 originally requires to sample link $Mm$ times because we need to search $M$ links that connected to $w_0$ in the first step and then sample one link for each of the $M$ nodes in each following step. A tree-structured sampling strategy may reduce this number: Specifically, we sample $k_i$ links for each node in step $i$ but we make sure $\prod_{i=1}^{m} k_i = M$, which does not change the total number of walks. In this way, the times of link search decrease to $\sum_{i=1}^{m} k_1 k_2 ... k_i$. Suppose $M = 64$, $m = 3$, and $k_1 = 4, k_2 = 4, k_3 = 4$, then the times of link search decrease to about $0.44Mm$ . Though empirical results below show that tree-structured sampling achieves slightly worse performance, it provides an opportunity to tradeoff between prediction performance and time complexity.

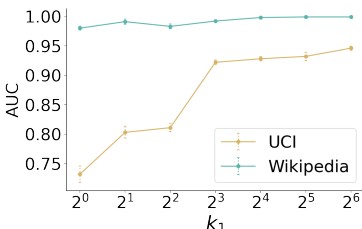

Figure 7: Effect of sampling of different tree structures on inductive performance.

We conduct more experiment to investigate this topic with Wikipedia and UCI datasets. The setup is as follows: first, we fix CAW sampling number $M = 64 = 2^6$ and length $m = 2$, so that we always have $k_1 k_2 = M = 2^6$; next, we assign different values to $k_1$, so that the shape of the tree changes accordingly; controlling other hyperparameters to be the optimal combination found by grid search, we plot the corresponding inductive AUC scores of CAW-N-mean on all testing edges in Fig. 7. It is observed that while tree-structured sampling may affect the performance to some extent, its negative impact is less prominent when the first-step sampling number $k_1$ is relatively large, and our model still achieves state-of-the-art performance compared to our baselines. That makes the tree-structured sampling a reasonable strategy that can further reduce time complexity.

## C    Additional Experimental Setup Details

### C.1    Dataset Introduction and Access

We list the introduction of the six datasets as follows.

- Reddit[1] is a dataset of posts made by users on subredditts over a month. Its nodes are users and posts, and its links are the timestamped posting requests.
- Wikipedia[2] is a dataset of edits over wiki pages over a month, whose nodes represent human editors and wiki pages and whose links represent timestamped edits.
- Social Evolution[3] is a dataset recording the detailed evolving physical proximity between students in a dormitory over a year, deterimined from wireless signals of their mobile devices.
- Enron[4] is a communication network whose links are email communication between core employees of a cooperation over several years.
- UCI[5] is a dataset recording online posts made by university students on a forum, but is non-attributed.
- MOOC[6] is a dataset of online courses where nodes represent students and course content units such as videos and problem sets, and links represent student's access behavior to a particular unit.

[1]http://snap.stanford.edu/jodie/reddit.csv
[2]http://snap.stanford.edu/jodie/wikipedia.csv
[3]http://realitycommons.media.mit.edu/socialevolution.html
[4]https://www.cs.cmu.edu/~./enron/
[5]http://konect.cc/networks/opsahl-ucforum/
[6]http://snap.stanford.edu/jodie/mooc.csv

## C.2 Baselines, Implementation and Training Details

### C.2.1 CAW-N-mean and CAW-N-attn

We first report the general training hyperparameters of our models in addition to those mentioned in the main text: on all datasets, we train both variants with mini-batch size 32 and set learning rate = $1.0 \times 10^{-4}$; the maximum training epoch number is 50 though in practice we observe that with early stopping we usually find the optimal epoch in fewer than 10 epochs; our early stopping strategy is that if the validation performance does not increase for more than 3 epoch then we stop and use the third previous epoch for testing; dropout layers with dropout probability = 0.1 are added to the RNN module, the MLP modules, and the self-attention pooling layer. Please refer to our code for more details.

In terms of the three hyperparameters controlling CAW sampling, we discussed them in Sec 5.3. For all datasets, they are systematically tuned with grid search, whose ranges are reported in Tab.4.

| Dataset | Sampling number $M$ | Time decay $\alpha$ | Walk length $m$ |
|---|---|---|---|
| Reddit | 32, 64, 128 | $\{0.25, 0.5, 1.0, 2.0, 4.0\} \times 10^{-5}$ | 1, 2, 3, 4 |
| Wikipedia | 32, 64, 128 | $\{0.25, 0.5, 1.0, 2.0, 4.0\} \times 10^{-6}$ | 2, 3, 4 |
| MOOC | 32, 64, 128 | $\{0.25, 0.5, 1.0, 2.0, 4.0\} \times 10^{-6}$ | 2, 3, 4, 5 |
| Social Evo. | 32, 64, 128 | $\{0.25, 0.5, 1.0, 2.0, 4.0, 8.0\} \times 10^{-6}$ | 1, 2, 3 |
| Enron | 32, 64, 128 | $\{0.25, 0.5, 1.0, 2.0, 4.0\} \times 10^{-7}$ | 1, 2, 3, 4 |
| UCI | 32, 64, 128 | $\{0.6, 0.8, 1.0, 1.2, 1.4\} \times 10^{-5}$ | 1, 2, 3 |

Table 4: Hyperparameter search range of CAW sampling.

Apart from the hyperparameters controlling CAW sampling, hidden dimensions of CAW-N mentioned in Sec.4.3, including that of the various encodings, MLPs, RNN, and attention projection matrices, are relatively less tuned. We select them based on two principles: 1) when node & link attributes are available, dimension of all these modules are set to have the same dimensions as baselines; 2) when node & link attributes are unavailable, the dimensions are picked from 32, 64, 128, whichever leads to a better performance.

### C.2.2 Baselines

We list the introduction of the six baselines as follows:

- DynAERNN (Goyal et al., 2020) uses a fully connected encoder to acquire network representations, passes them into LSTM and uses a fully connected network to decode the future network structures.
- JODIE (Kumar et al., 2019) applies RNNs to estimate the future embedding of nodes. The model was proposed for bipartite graphs while we properly modify it for standard graphs if the input graphs are non-bipartite.
- DyRep (Trivedi et al., 2019) also uses RNNs to learn node embedding while its loss function is built upon temporal point process.
- VGRNN (Hajiramezanali et al., 2019) generalizes the variational GAE (Kipf & Welling (2016)) to temporal graphs, which makes the prior depend on the historical dynamics and captures those dynamics with RNNs.
- EvolveGCN (Pareja et al., 2020) uses a RNN to estimate the GCN parameters for the future snapshots.
- TGAT (Xu et al., 2020) leverages GAT to extract node representations where the nodes' neighbors are sampled from the history and encodes temporal information via Eq.7.

We introduce how we tune these baselines as follows.

**DynAERNN**. The model with code provided **here** is adapted into our evaluation pipeline. We follow most of the settings in the code. We tune the embedding size in {32, 64} and lookback in {2, 3, 5} to report the best performance.

**JODIE**. The model with code provided **here** is adapted into our evaluation pipeline. JODIE calculates the $L_2$ distances between the predicted item embedding to other items and uses the rankings to evaluate their performance. Here, we consider the negative distances as the prediction score. Based on the prediction score, we calculate mAP and AUC. We split the data according to the setting in section 5.1. The model is trained for 50 epoches. The dimensions of the dynamic embedding is searched in{64, 128} and the best performance is reported.

**DyRep**. The model with code provided **here** is adapted into our evaluation pipeline. We follow most of the settings in the paper. That is, we set the number of samples for survival to 5, gradient clipping to 100. And we tune the hidden unit size and embedding size in {32, 64} to report the best performance. The model uses likelihood based on point process to predict links and therefore we use these likelihood scores to compute AUC and AP.

**VGRNN**. The model with code provided **here** is adapted into our evaluation pipeline. We use several of its default settings: one-hot node features as input when node attributes are unavailable as suggested by the original paper (Hajiramezanali et al., 2019), one layer of GRU network as the history tracking backbone, a learning rate of 1e-2, and training for 1000 epochs. Its hidden dimension is searched in {32, 64} and the best performance is reported.

**EvolveGCN**. The model with code provided **here** is adapted into our evaluation pipeline. We utilize EvolveGCN-O version since it can capture more graph structural information. For most hyperparameters, we follow the default values. According to our setting, we sample an equal amount of negative links, which means we set negative_mult_training and negative_mult_test to 1. One central hyperparameter needs to be further tuned is number of previous snapshots used for training and testing. We search its optimal value in {3, 4, 6, 8, 10} when tuning the model for most datasets. Since Enron only contains 11 snapshots, we search its optimal value in {3,4,5,6}.

**TGAT**. The model with code provided **here** is adapted into our evaluation pipeline. We use several of their default settings. That is, we use product attention, set the number of attention heads to 2, set the number of graph attention layers to 2, and use 100 as their default hidden dimension. One central hyperparameter that needs to be further tuned is the degree of their neighbor sampling. We search its optimal value in {10, 20, 30} when tuning the model.

### C.3 EVALUATION OF SNAPSHOT-BASED BASELINES

| | Reddit | Wikipedia | MOOC | Social Evo. | Enron | UCI |
|---|---|---|---|---|---|---|
| total snapshots | 174 | 20 | 20 | 27 | 11 | 88 |
| exact split | 122 / 26 / 26 | 14 / 3 / 3 | 14 / 3 / 3 | 19 / 4 / 4 | 7 / 2 / 2 | 62 / 13 / 13 |
| referenced baseline | EvolveGCN | - | - | VGRNN | VGRNN | EvolveGCN |

Table 5: Snapshot split for evaluating snapshot-based baselines.

We make the following decisions to evaluate snapshot-based baselines in a fair manner, so that their performances are comparable to those derived from the stream-based evaluation procedure. The first step we do is to evenly split the whole dataset chronologically into a number of snapshots. We determine the exact number of snapshots by referring the three snapshot-based baselines we use. For Wikipedia and MOOC dataset which are not used by any snapshot-based baseline, we split them into a total of 20 snapshots. Next, we need to determine the proportions of these snapshots assigned each to training, validation, and testing set. In doing this, our principle is that the proportions of these three sets should be close to 70:15:15 as much as possible, since that ratio is what we use for evaluating stream-based baselines and our proposed method. These decisions lead to our final splitting scheme summarized in Tab. 5.

Extra care should also be taken when testing snapshot-based methods. For a queried link in a snapshot, usually snapshot-based methods only make a binary prediction whether or not that link may exist at any time in that snapshot. They do not, however, take care of the case that the link may appear multiple times at different time points within that snapshot's time range. This lead to a different evaluation scheme than stream-based methods, which do consider the multiplicity of links. Therefore,

when testing snapshot-based methods, if a link appear in a certain snapshot for multiple times, we record the model's prediction the same number of times before computing its performance metrics.

## C.4 Choice of Evaluation Metric

When considering link prediction as a binary classification problem, the existing literature usually choose metrics from the following: Area Under the ROC Curve (AUC), Average Precision (AP), and Accuracy (ACC). The reason we do not use ACC is that a proper confidence threshold of decision is ill-defined in literature, which leads to unfair comparison across different works.

## C.5 Computing Infrastructure

All the experiments were carried out on a Ubuntu 16.04 server with Xeon Gold 6148 2.4 GHz 40-core CPU, Nvidia 2080 Ti RTX 11GB GPU, and 768 GB memory.

# D  Additional Experimental Results

## D.1 Performance in Average Precision

| Task | | Methods | Reddit | Wikipedia | MOOC | Social Evo. | Enron | UCI |
|---|---|---|---|---|---|---|---|---|
| Inductive | new v.s. new | DynAERNN | $58.63 \pm 5.42$ | $54.94 \pm 2.29$ | $59.84 \pm 1.26$ | $54.76 \pm 1.33$ | $54.89 \pm 3.79$ | $51.59 \pm 3.92$ |
| | | JODIE | $80.03 \pm 0.13$ | $76.90 \pm 0.49$ | $82.27 \pm 0.46^\dagger$ | $87.96 \pm 0.12^\dagger$ | $79.80 \pm 1.48^\dagger$ | $71.64 \pm 0.62$ |
| | | DyRep | $61.28 \pm 1.89$ | $57.57 \pm 2.56$ | $62.29 \pm 2.09$ | $75.42 \pm 0.32$ | $69.97 \pm 0.92$ | $63.08 \pm 7.40$ |
| | | VGRNN | $60.64 \pm 0.68$ | $52.55 \pm 0.82$ | $65.44 \pm 0.82$ | $67.83 \pm 0.53$ | $67.93 \pm 0.88$ | $67.50 \pm 0.92$ |
| | | EvolveGCN | $62.99 \pm 0.17$ | $55.64 \pm 1.03$ | $52.28 \pm 1.80$ | $52.26 \pm 1.16$ | $47.36 \pm 1.24$ | $80.98 \pm 1.09^\dagger$ |
| | | TGAT | $95.17 \pm 0.91^\dagger$ | $93.18 \pm 0.73^\dagger$ | $72.91 \pm 0.92$ | $52.17 \pm 1.94$ | $63.83 \pm 3.70$ | $75.27 \pm 2.34$ |
| | | **CAW-N-mean** | $\mathbf{98.05 \pm 0.87}^*$ | $\mathbf{96.01 \pm 0.25}^*$ | $\mathbf{90.36 \pm 0.80}^*$ | $\mathbf{92.16 \pm 1.03}^*$ | $\mathbf{93.93 \pm 0.66}^*$ | $\mathbf{99.63 \pm 0.34}^*$ |
| | | **CAW-N-attn** | $\mathbf{98.08 \pm 0.66}^*$ | $\mathbf{97.86 \pm 0.63}^*$ | $\mathbf{90.35 \pm 0.81}^*$ | $\mathbf{93.29 \pm 1.90}^*$ | $\mathbf{92.73 \pm 0.76}^*$ | $\mathbf{100.00 \pm 0.00}^*$ |
| | new v.s. old | DynAERNN | $66.59 \pm 2.90$ | $63.76 \pm 2.82$ | $82.02 \pm 1.59$ | $52.54 \pm 0.22$ | $55.50 \pm 2.07$ | $57.29 \pm 2.52$ |
| | | JODIE | $83.15 \pm 0.03$ | $80.54 \pm 0.06$ | $87.95 \pm 0.08$ | $91.40 \pm 0.04^\dagger$ | $89.57 \pm 0.30$ | $76.34 \pm 0.17$ |
| | | DyRep | $66.73 \pm 1.99$ | $76.89 \pm 0.31$ | $88.25 \pm 1.20^\dagger$ | $89.41 \pm 0.29$ | $95.97 \pm 0.28^\dagger$ | $93.60 \pm 1.47^\dagger$ |
| | | VGRNN | $52.84 \pm 0.66$ | $60.99 \pm 0.55$ | $62.95 \pm 0.58$ | $69.20 \pm 0.52$ | $67.93 \pm 0.88$ | $67.50 \pm 0.92$ |
| | | EvolveGCN | $66.29 \pm 0.52$ | $53.82 \pm 1.64$ | $51.53 \pm 0.92$ | $52.01 \pm 0.67$ | $46.56 \pm 1.89$ | $76.30 \pm 0.33$ |
| | | TGAT | $97.09 \pm 0.18^\dagger$ | $95.17 \pm 0.15^\dagger$ | $71.77 \pm 0.23$ | $52.48 \pm 0.52$ | $59.70 \pm 1.49$ | $75.01 \pm 0.72$ |
| | | **CAW-N-mean** | $\mathbf{98.89 \pm 0.04}^*$ | $\mathbf{99.04 \pm 0.04}^*$ | $\mathbf{90.99 \pm 0.96}^*$ | $\mathbf{93.71 \pm 0.75}^*$ | $92.93 \pm 0.94$ | $\mathbf{98.86 \pm 0.95}^*$ |
| | | **CAW-N-attn** | $\mathbf{99.90 \pm 0.05}^*$ | $\mathbf{99.55 \pm 0.30}^*$ | $\mathbf{91.24 \pm 0.93}^*$ | $\mathbf{94.17 \pm 0.86}^*$ | $92.38 \pm 1.36$ | $\mathbf{98.53 \pm 0.64}^*$ |
| Transductive | | DynAERNN | $85.58 \pm 2.12$ | $76.58 \pm 1.41$ | $89.29 \pm 0.49$ | $66.58 \pm 1.84$ | $60.90 \pm 2.70$ | $84.95 \pm 2.13$ |
| | | JODIE | $91.14 \pm 0.01$ | $91.39 \pm 0.04$ | $91.19 \pm 0.03^\dagger$ | $89.22 \pm 0.01$ | $91.94 \pm 0.01$ | $80.27 \pm 0.08$ |
| | | DyRep | $67.54 \pm 2.02$ | $77.36 \pm 0.25$ | $90.49 \pm 0.03$ | $94.48 \pm 0.01^\dagger$ | $97.14 \pm 0.07^\dagger$ | $95.29 \pm 0.13^\dagger$ |
| | | VGRNN | $50.87 \pm 0.81$ | $67.66 \pm 0.89$ | $83.70 \pm 0.56$ | $78.66 \pm 0.67$ | $94.02 \pm 0.52$ | $82.23 \pm 0.56$ |
| | | EvolveGCN | $54.49 \pm 0.73$ | $55.84 \pm 0.37$ | $51.80 \pm 0.46$ | $56.90 \pm 0.54$ | $69.72 \pm 0.49$ | $81.63 \pm 0.23$ |
| | | TGAT | $98.38 \pm 0.01^\dagger$ | $96.65 \pm 0.06^\dagger$ | $69.75 \pm 0.23$ | $57.37 \pm 1.18$ | $57.37 \pm 0.18$ | $60.25 \pm 0.31$ |
| | | **CAW-N-mean** | $\mathbf{99.96 \pm 0.01}^*$ | $\mathbf{99.91 \pm 0.03}^*$ | $\mathbf{92.05 \pm 0.88}$ | $\mathbf{95.90 \pm 0.09}^*$ | $94.93 \pm 0.39$ | $\mathbf{95.89 \pm 0.87}$ |
| | | **CAW-N-attn** | $\mathbf{99.99 \pm 0.01}^*$ | $\mathbf{99.89 \pm 0.03}^*$ | $\mathbf{92.23 \pm 0.76}^*$ | $\mathbf{96.37 \pm 0.09}^*$ | $96.13 \pm 0.37$ | $\mathbf{98.86 \pm 0.38}^*$ |

Table 6: Performance in Average Precision (AP) (mean in percentage $\pm$ 95% confidence level.) $\dagger$ highlights the best baselines. *, **bold font**, **bold font**$^*$ respectively highlights the case where our models' performance exceeds the best baseline on average, by 70% confidence, by 95% confidence.

## D.2 More Discussion on Stream-based vs. Snapshot-based Methods

Stream-based methods usually treat each temporal link as an individual training instance. In contrast, snapshot-based methods stack all the temporal links within a time slice into one static graph snapshot and do not further distinguish temporal order of links within that snapshot. In Tab. 2 and 6 we saw that stream-based methods generally exhibit better performance than snapshot-based methods. An important reason is that stream-based methods are able to access the few most recent interactions previous to the target link to predict. This makes them especially advantageous when used to model many common temporal networks whose dynamics are governed by some short-term laws. In contrast, snapshot-based methods are less able to access such immediate history, because they make prediction on all links within one future snapshot all at once. In principle they could alleviate this problem by making very fine-grained snapshots so that less immediate history is missed. However, this is not practical with real-word large temporal graphs whose temporal links come in millions, which leads to snapshot sequences of extreme length intractable to their recurrent neural structure. This issue was

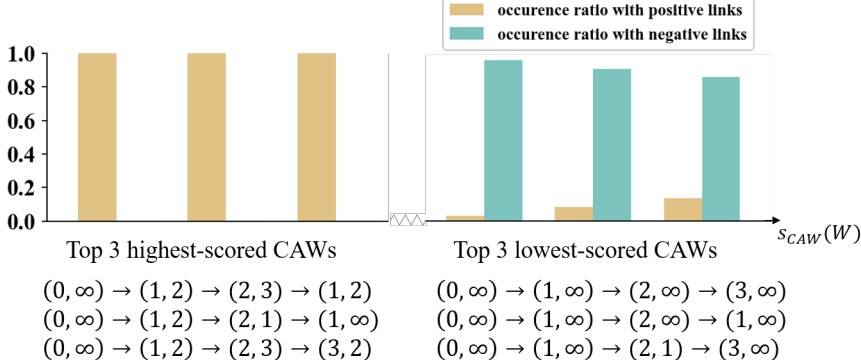

$$(0,\infty) \to (1,2) \to (2,3) \to (1,2)$$
$$(0,\infty) \to (1,2) \to (2,1) \to (1,\infty)$$
$$(0,\infty) \to (1,2) \to (2,3) \to (3,2)$$

$$(0,\infty) \to (1,\infty) \to (2,\infty) \to (3,\infty)$$
$$(0,\infty) \to (1,\infty) \to (2,\infty) \to (1,\infty)$$
$$(0,\infty) \to (1,\infty) \to (2,1) \to (3,\infty)$$

Figure 8: Visualizing most discriminatory CAWs, and their occurrence ratios with positive / negative samples.

also observed by the recent work to model social interacting behaviors (Wang et al., 2020), although temporal convolutional networks may alleviate this issue to some extent. That said, snapshot-based methods usually has the advantage that they usually consume less memory and computation time. Stream-based methods on the other hand need to manage how they sample history very carefully to balance the efficiency and effectiveness. Our proposed algorithm on CAW sampling comes into the place in light of this to solve the problem.

## E    VISUALIZING CAWS AND AWS

Here, we introduce one way to visualize and interpret CAWs. Our interpretation can also illustrate the importance to capture the correlation between walks to represent the dynamics of temporal networks, where the set-based anonymization of CAWs can work while AWs will fail. The basic idea of our intrepretation is to identify different shapes of CAWs via their patterns encoded in $I_{CAW}$, and compare their contributions to the link-prediction confidence. The idea will be also used to interpret AWs so that we can compare CAWs with AWs.

First, we define the shapes of walks based on $I_{CAW}$. Recall from Eq. 3 that $g(w, S_u)$ encodes the number of times node $w$ that appears in different walk positions w.r.t source node $u$. This encoding induces a temporal shortest-path distance $d_{uw}$ between node $w$ and $u$: $d_{uw} \triangleq \min\{i | g(w, S_u)[i] > 0\}$. Note that in temporal networks, there is not a canonical way to define shortest-path distance between two nodes as there is no static structures. So our definition $d_{uw}$ can be viewed the shortest-path distance between u and w over the subgraph that consists of walks in $S_u$. Based on the way to define $(d_{uw}, d_{vw})$, we introduce the mapping from $I_{CAW}$ of the node $w$ to a coordinate of this node in the subgraph that consists of walks in $S_u \cap S_v$: $(g(w, S_u), g(w, S_v)) \to \text{coor}(w; u, v) = (d_{uw}, d_{vw})$. This cooredinate can be viewed as a relative coordinate of node $w$ w.r.t. the source nodes $u$, $v$. Each walk $W \in S_u \cup S_v$ can then represented as a sequence of such coordinates by mapping each node's $I_{CAW}$ to a coordinate. The obtained sequence can be viewed as a shape of $W$ and we denote the obtained shape as $s_{CAW}(W)$. For instance, in the toy example shown by Fig. 2 right, the first CAW in $S_u$, $u \to b \to a \to c$ is mapped to a new coordinate sequence $(0,2) \to (1,2) \to (2,\infty) \to (3,\infty)$. The $\infty$ marks the setting that node $a$, $c$ do not appear in $S_v$.

Next, we score the contributions of CAWs with different shapes. We use CAW-N-Mean as $\text{enc}(S_u \cup S_v)$ is simply mean over the encodings of sampled CAWs and further use linear projection $\beta^T \text{enc}(S_u \cup S_v)$ to compute the final scalar logit for prediction. As the two operations mean and projection are commutative, the above setting allows each CAW $\hat{W}_i$ contributing a scalar score $logit(\hat{W}_i) = \beta^T \text{enc}(\hat{W}_i)$ to the final logit.

Fig. 8 lists the 3 highest-scored and 3 lowest-scored shapes of CAW, which are extracted from the Wikipedia dataset with $M = 32$ and $m = 3$. A law of general motif closure can be observed from the highest-scored CAWs: two nodes that commonly appear in some types of motif are more inclined to have a link in between. For example, the highest-scored shape of CAW, $(0,\infty) \to (1,2) \to (2,3) \to (1,2)$, implies that the nodes except the first in this CAW appear in the sampled common 3-hop

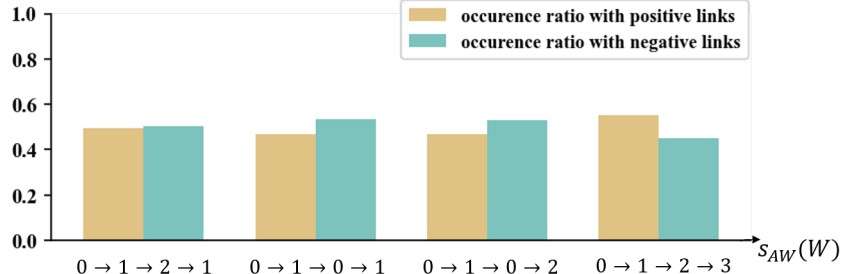

Figure 9: Visualizing *all* AWs, and their occurrence ratios with positive / negative samples.

neighborhood around the two nodes between which the link is to be predicted. Therefore, CAW-N essentially adaptively samples a temporal motif closure pattern that is very informative to predict this link. CAW-N does not explicitly enumerating or counting these motif patterns. In contrast, when CAWs do not bridge the two nodes, as shown in top-2 lowest-scored CAWs, very unlikely there will exist a link. Fig. 8 also displays each of the 6 CAW's occurrence ratio with positive and negative links. The difference within each pair of ratios is an indicator of the corresponding CAW's discriminatory power. We also see that the discriminatory power of CAWs is very strong: highest-scored CAWs almost never occur with negative links, and lowest-scored CAWs also seldom occur with positive links.

We further apply the similar procedure to analyze the AWs introduced in Sec. 4.1 and the model Ab.5 of Tab. 3 used for ablation study. Note that AW cannot establish the correlation between walks, each single AW itself, say $W = (v_0, v_1, ..., v_m)$, decides its own shape $s_{AW}(W)$. We directly set $s_{AW}(W) = I_{AW}(v_0; W) \to I_{AW}(v_2; W) \to \cdots \to I_{AW}(v_m; W)$ with $I_{AW}(w; W)$ defined in Eq. 2. As the Wikipedia dataset is a bipartite graph, there are in total four different shapes when $m = 3$ as listed align with the x-axis of Fig. 9. For illustration, we explain one shape of AW as an example, say $0 \to 1 \to 2 \to 1$: The corresponding walks have the second and the fourth nodes correspond to the same node, which the first, second and third nodes are different. As shown in Fig. 9, we can see that AW's occurrence with positive versus negative links are highly mixed-up, compared to CAW's. That suggests that AWs possess significantly less discriminatory power than CAWs. The main reason is that AWs do not have set-based anonymization, so they cannot capture the correlation between walks/motifs but CAWs can do that. This observation further gives a reason on why the model Ab.5 of Tab. 3 only achieves some performance on-par with the model Ab.2 where we totally remove the anonymization procedure: the anonymization adopted by AWs loses too much information of the structure and cannot benefit the prediction much. However, the original CAW-N well captures such information via the set-based anonymization.

