# OpenReview forum: "Inductive Representation Learning in Temporal Networks via Causal Anonymous Walks"
_ICLR.cc/2021/Conference — ICLR 2021 Poster_

### Official Review · AnonReviewer2 · 2020-10-28
**Confusion about the notion of inductiveness**

**Rating:** 7
**Confidence:** 4

**Review:**

The authors provide in-depth analysis on the critical topic of capturing dynamic laws for the inductive representation learning of temporal graphs. The authors leverage the causal anonymous walk to capture the topological laws of the dynamic graph, while not requiring to memorize node identities such that inductive learning is still feasible. Compared with the previous work, the paper emphasizes on scenarios where node/edge attributes are less informative of the inductive reasoning. Some workaround methods are also proposed to deal with the edge features and to enable the efficient computation and sampling procedure.

The paper is overall well-written, and the motivations are justified clearly. The most relevant work of TGAT from Xu, et al. 2020 does not explicitly account for the dynamic laws of evolving graphs, and the way they construct and sample the local neighborhood is infeasible when processing large graphs in a streaming fashion. The authors leverage the causal anonymous walks to:

1. capture the signals of topological structures which "reflect" the identities of the nodes;

2. avoid the costly computations to recover the network structural features during both the training and inference.

The contribution is solid as it complements the TGAT when the node/edges features are less important. It also solves the computation issues of the TGAT. The numerical results also compare favorably to the baselines.

My concerns with the paper include:

1. Using random walks and local aggregation methods (such as GCN, GAT, TGAT) have their different emphasis on the network topological laws & node/edge features. The authors should discuss and make it more clear regarding the pros and cons.

2. It appears TGAT adopts the same inductive representation learning setting from GraphSAGE and GAT, which are widely accepted in this domain. The claim and key arguments made by the authors raise a major confusion about being inductive in graph representation learning, and I believe the criticisms here are equally applicable to GraphSAGE and GAT that leverage node features as well. Therefore, I think the notion of inductiveness studied by the authors is at most in parallel to that of TGAT (and GraphSAGE & GAT). If the authors believe their notion of inductiveness is more proper, they should discuss the similar issues for GraphSAGE and GAT since they are more well-known to the readers and will raise less confusion.

3. the link sampling strategy that ensures the overall efficiency of the computation lacks support and is not principled. To some extent, neighborhood/walk construction in dynamic graphs are still open challenges in this domain, so I think using heuristics is fine, but the consequence should be made more clear.

The authors make the novel contribution of combining the causal anonymous walk with inductive representation on temporal graphs. However, their arguments on the notion of inductiveness can raise a major confusion if not directly discussed for GraphSAGE and GAT. Therefore, I vote for a weak acceptance of this version of the paper.

============================================
The authors address most of my questions and concerns in the response, and update the manuscript. I decide to increase my score.

---

> ### Author Response · Authors · 2020-11-17
> **Clarification of the notion of inductiveness and the link sampling strategy**
>
> We thank R2 for the valuable comments on our paper. We especially thank R2 for clearly positioning the contributions of our approach, insightfully asking very fundamental questions. We address these questions as the follows.
>
> ## Q1. "The comparison between random walks and local aggregation methods"
> * We agree the statement made by R2. In general, they have different emphasis. GCN, GAT, TGAT favor the case with very informative node features and node features smoothing. However, random walk methods more favor network dynamics representation. Our goal is to extract the network dynamics and therefore we adopt the more suitable random walk methods. Note that although most previous works on representation learning of temporal networks also focused on extracting the network dynamics (link prediction), they did not choose such more suitable (random walk) methods. We see this as one of our contributions.
> * More importantly, random-walk methods are also naturally suitable for the computation of relative node identities, which is critical to encode the network dynamics when node identities are removed and no informative node/edge attributes are available.
>
> ## Q2. "The notion of inductiveness"
> * Regarding the notion of inductiveness, we are sorry for the caused confusion and have fixed our statement in the revised manuscript. Actually, in this paper, we still use the widely accepted definition of inductiveness in machine learning. That is, an inductive model can be applied to a testing dataset whose labels and features can never be used for training. Regarding this notion, we should say TGAT, GraphSAGE and GAT are all inductive. We can see in Table 2, the performance gap of TGAT between the inductive setting and the transductive setting is not huge. Therefore, TGAT is inductive. We fixed our argument in the revised manuscript.
> * However, the issue of TGAT is that it only performs well (even just in training datasets) when informative node features are available. GraphSAGE and GAT share the same issue. Therefore, in the section 2, we explicitly discuss the condition when TGAT fails, i.e., when informative node/edge attributes are not available. We also criticize GraphSAGE and GAT who will fail in a similar case.
> *Our CAW-N does not depend on informative node features to represent the network dynamics. Some interpretation and visualization results are also provided in Appendix E of the revised manuscript, where we illustrate some patterns of CAWs to indicate how CAW-N can work so well to represent network dynamics.
>
> ## Q3. "The link sampling strategy is not principled."
> * Regarding the argument that the link sampling strategy is just heuristic, we respectfully disagree. First, we argue that the link sampling strategy is not related whether the method is random walk or local aggregation. Therefore, it is an independent contribution and can also be used to improve local aggregation.
> * In appendix A, we have rigorous demonstration on why our sampling strategy does not need to record the entire history. Actually, when the entire history has to be recorded, as TGAT did, the memory complexity is O(|E|) and the time complexity is O(|E|^2). In practice, |E| increases with respect to time as new edges arrive. This may yield some practical issues of TGAT. In contrast, our approach does not need to record the entire history. Therefore, if the edges come in as a stream, our model only needs to record at most constant many recent edges for each node pair. The memory complexity will not increase when new edges come in and the time complexity of our model is O(|E|). We demonstrate the linear time complexity of our approach in Sec. 5.4.

---

> > ### Comment · AnonReviewer2 · 2020-11-24
> > **Thanks for the clarification**
> >
> > I want to thank the authors for the response and updating the manuscript. The updated version addresses some of my concerns. I decide to vote for an acceptance.

---

> > > ### Author Response · Authors · 2020-11-24
> > > **Thanks for your vote**
> > >
> > > The authors would like to thank R2 for appreciating this work and voting for an acceptance.

---

### Official Review · AnonReviewer1 · 2020-10-29
**Proposing an anonymous walk methodology to support link prediction in temporal networks**

**Rating:** 6
**Confidence:** 3

**Review:**

The proposed causal anonymous walks (CAW) is the key contribution of the paper where the set-based anonymization is adopted to allow temporal and structure information in a temporal network can be retained in the CAWs. The sampled CAWs are then used to learn RNN for the link prediction.

Pros:
- The proposed design of CAW (set-based anonymization in particular) is a simple and reasonable one. The empirical results show that it is a good input representation to be fed to the RNN for training.
- The consideration of the underlying motifs is captured by the RNN so that the scalability issue caused by identifying the motifs can be alleviated.
- The empirical results based on six different datasets show that it outperforms a number of existing methods.
- The experiments were well designed with ablation study to figure out the effectiveness of different components introduced in the proposed methods.

Cons:
- In terms of methodology, it is an improved method upon some previously proposed anonymous walk for handling temporal networks. There is some novelty but it is hard to be considered as a breakthrough as claimed by the authors.
- Figures 1 and 2 are included to help explain the concepts. They are good in general but a bit compact with a lot of details packed making the presentation not hard to comprehend.
- Readability could be improved. My impression is that it requires the readers to have some background on temporal motifs and anonymous walks or it could be hard for the reader to follow.
- Entanglement of structural and temporal information is mentioned as one of the challenges. It is not clear how this is addressed in the proposed method.

Qns
- Is the structural and temporal information being addressed disentangled in some way? It seems that it just replies on the use of RNN to take care of all the information as a black box.
- Can the optimality of the adopted encoding steps be discussed? It seems that they are determined by making references to different existing works.

---

> ### Author Response · Authors · 2020-11-17
> **Clarification of the significance of CAW-N and how we deal with structural and temporal information**
>
> We thank R1 for the valuable comments on our paper. We especially thank R 1 for appreciating the novelty of our approach and our experiments. We address the concerns raised by R1 as the follows.
>
> ## Q1. "The significance of the model vs the AW model"
> * Regarding the significance, R1 seems to miss capturing the function of the set-based anonymization used in CAW that is significantly different from the anonymization procedure in AW, and thus incorrectly overlooked the significance of our CAW. CAW only shares a high-level similar idea with the AW approach but it works in a substantially different mechanism.
>
> * First, the problems that two methods (CAW and AW) aim to solve are significantly different. The previous AW approach was used for entire graph representation while our CAW approach is used to perform link prediction. The problem setting itself has been already more than just an extension from a static setting to a temporal setting. The difference in problem setting also leads to the fundamental difference between CAW’s mechanism and AW’s one.
>
> * Second, CAW can model the correlation of motifs, which is crucial to learn the network dynamics but AW cannot. In the manuscript, we have introduced two types of technical differences between CAWs and AWs. One is the way to extract walks, where CAWs use temporal random walk to encode causality. We would have agreed with the review 1 if we simply generalized AWs by performing temporal random walk and kept the remaining steps in the same way, which is a direct extension of AW by injecting temporal information into the model. However, we did much more than that. The above direct extension does not work well, as shown in our ablation study No. 5 in Table 3. This observation motivated us to propose the more significant contribution, the set-based anonymization adopted by CAW, which can model the correlation between the walks. AW-type anonymization will miss the correlation. Therefore, AWs hold the same issue as T-GAT and fail in the example shown in Fig. 3. But CAW-type anonymization is the key to solve that issue. We also provide visualization results in Appendix E to illustrate why the correlation captured by CAWs is very crucial.
>
> * To the best of our knowledge, there has been no previous works proposing this idea, even just similar in the high level, to establish the correlation between walks/motifs. Our anonymization is simple, effective and extremely novel. Therefore, CAWs are significant improvement beyond AWs.
>
> ## Q2. "How does the model handle the entanglement of structural and temporal information?"
> * Regarding the entanglement of structural and temporal information, we would like to further clarify this concept. The ideal solution to handle such a challenge is not to literally process structural and temporal information separately, as imagined by the reviewer 1. Essentially, these two sides of the information are mutually interacted. If we process them separately, we may lose useful information. Because these two sides need to be processed simultaneously and their correlation should be properly captured, the model design in general becomes very hard. Previous tools are either for structural information (such as GNN) or temporal information (such as RNN). Therefore, researchers must reduce such a complicated data structure into one of the above two cases.
> * Our approach, however, provides an elegant way to model both-side information. Specifically, we adopt temporal random walks that involve both structural and temporal information. The obtained walks reduce the data structure into a sequential format. Then, we use set-based anonymization to further inject structural information and further use RNNs to enhance temporal information. The design we adopted essentially injects structural and temporal information in an iterative manner. The exceptionally well experimental results demonstrate the effectiveness of our model to deal with the entanglement of structural and temporal information.
>
> ## Q3. "The details on the optimality of time encoding"
> The capability of Fourier features (adopted as the time encoding) to approximate any positive definite kernels seems to be a well-established result in statistical learning theory. As it is a direct result from previous works and due to the page limitation, we omit the detailed discussion on it and refers the previous related references.

---

### Official Review · AnonReviewer4 · 2020-10-29
**Official Blind Review #4**

**Rating:** 6
**Confidence:** 4

**Review:**

Summary:
The paper provides an interesting way to inductively represent a temporal network with the proposed Causal Anonymous Walks (CAWs) which work as temporal motifs to represent the network dynamics. The CAWs can be further encoded by the proposed CAW-N which supports online training.

Pros:
Overall, I like the idea to represent the temporal network with the proposed causal anonymous walks. Experimental results also show significant improvements over the existing baselines.
- Representation learning for dynamic graphs is very practical and important. This paper proposed an effective solution to this problem.
- The proposed CAW is novel for capturing the temporal dynamics in temporal networks. The description of the method is easy to follow.
- This paper provides sufficient experimental results which show the effectiveness of the proposed CAW-N model on the link prediction task, including a wide variety of baselines and datasets.

Cons:
- This paper has been listed on the author's homepage (https://scholar.google.com/citations?user=Ch3YUgsAAAAJ&hl=en), potentially violate the double blind review rules.
- Some figures, e.g. figure 4, are not very useful in explaining the corner cases. Simple sentences are enough for clarification.
- It would be better to include the training/inference time comparisons as the introduction claims "the model scalability".
- It would be more convincing if the authors can provide representation visualization comparisons in the rebuttal period.

Minors:
- missing conclusion section
- Eq.2, I_AW(w; P), definition of P

---

> ### Author Response · Authors · 2020-11-17
> **Adding complexity evaluation of CAW-N and visualization of what CAW-N has learnt**
>
> We thank R4 for the valuable comments on our paper. We especially thank R4 for appreciating the novelty of our approach and our experiments. We address the concerns as the follows.
> ## Q1. “The study of the model scalability”
> * We provide the time complexity evaluation on our method in Sec. 5.4 in the revised manuscript, which demonstrates the statement that the time complexity of our model only linearly depend on the number of edges. Our model only needs constant time to sample the history for each edge, which is due to our link sample strategy in the appendix A. Our llink sample strategy is solid and the key to significantly decrease the time complexity from O(|E|^2) to O(|E|) in total, where |E| is the number of edges.
>
> ## Q2. “Visualization of the representations/patterns learnt by the model”
> * Essentially, our method does not explicitly compute node representations. Therefore, we cannot provide the traditional node representation visualization. However, we may provide visualization on how CAW-N extracts the patterns to make successful link prediction. We put such results and analysis in Appendix E. The results show that CAW-N uses a law of motif closure to predict links.
>
> ## Q3. "Whether the authors break the double blind rules"
> * Regarding the double blind rules for ICLR (https://iclr.cc/Conferences/2021/CallForPapers), we do not think that the behavior of one of the authors' behavior (list the paper in his google scholar page) breaks the rules. The rules only ask authors to keep anonymous all through the paper, which we strictly followed. To avoid further disclosing the authors’ identities, the author who lists the paper in his google scholar page has removed the paper from the list.
>
> ## Q4. Other minor comments
> * We have provided a conclusion section in Sec. 6 and highlights some promising future research directions based on CAW-N. Regarding fig. 4, we would like to keep it to make the paper more accessibility to the general audience that are not familiar with the setting.

---

### Official Review · AnonReviewer3 · 2020-10-29
**INDUCTIVE REPRESENTATION LEARNING IN TEMPO- RAL NETWORKS VIA CAUSAL ANONYMOUS WALKS**

**Rating:** 5
**Confidence:** 4

**Review:**

This paper proposes Causal Anonymous Walks (CAWs) that are extracted by temporal random walks and work as automatic retrieval of temporal network motifs to represent network dynamics while avoiding the time-consuming selection and counting of those motifs. CAWs adopt an anonymization strategy that replaces node identities with the hitting counts of the nodes based on a set of sampled walks to keep the method inductive, and simultaneously establish the correlation between motifs. CAW-N to encode CAWs with a neural network model. CAW-N was evaluated to predict links over 6 real temporal showed better AUC gain compared to baselines.
Pros:
-	Promising evaluation results which shows average 15% AUC gain compared to 6 baselines over 6 real world datasets.
-	Novel idea of avoiding the selection and counting of network motifs instead leveraging anonymization and encoding.
Cons:
-	No conclusion or future work that discusses about next steps for research community.
-	Hyperparameter study shows the AUC is very sensitive to sampling strategy (time bias or sampling length).
-	Set-based anonymization seems to require a computational complexity of counting the number of nodes in different positions. For example in Figure 2, it counts the number of “b” in different positions. There is no computational complexity comparison between counting this vs. motif counting.

---

> ### Author Response · Authors · 2020-11-17
> **CAW-N is much more efficient than motif counting and robust to hyperparameters**
>
> We thank R3 for the valuable comments on our paper. We specially thank R3 for notifying us of the missing conclusion and discussion section. We have provided the conclusion section in the revised manuscript.
>
> Regarding other concerns raised by the reviewer 3, however, we respectfully disagree.
>
> ## Q1. “Complexity of our counting vs motif counting”
> **Our counting the numbers of nodes in different positions is far more efficient than counting temporal motifs.**
> * Note that for each edge used for training, the counting number in our case is in total 2Mm, where M is the number of walks for a node while m is the length of a walk. Both M and m are very small constants: In our experiments, the setting M=32 and m=3 in general achieves very good performance, though increasing M may slightly increase the performance. Our model is evaluated when M<=128 and m<=4 for all datasets.
>
> * However, counting temporal motif is much more complex. For every edge of interest, it needs to first track all the edges in the subgraph around this edge that appear in some predetermined time interval; Second it needs to solves graph isomorphism testing to decide the pattern of each motif that consist of the edges extracted in the previous step. The first step in general needs to track k edges, where k is hundreds or even thousands. Now we count the motifs: If we consider all the motifs with 2m edges (actually in our method, CAWs implicitly model motifs with up-to 2m edges, as CAWs establish the correlation between the walks of two nodes in a link), the related motif counting complexity is about O(k^{2m}). If we choose k=100, m=2, the complexity is about 10^8. Here, we even have not considered the complexity of solving the graph isomorphism problem for each motif.
> * To demonstrate the above arguement, we check the Wikipedia dataset: For an edge (u,v,t), we collect all the edges that appear within 2-hop neighborhood of either u or t and within time [t – 0.05T, t], where T is the total time range. Then, we count the motifs that contain 4 edges and nodes u, v. The total number of these motifs for an edge (u,v,t) are 10^6 in average and 10^8 in the worst case. It is significantly larger than 2Mm(<10^3) which is the counting complexity in our method.
> *Note that the motif counting method in [1] cannot be used here. Because we not only need to count the total number of motifs in the networks (the setting in [1]) but also count the motifs that contain a certain edge. That is, we need to locate each motif in the network multiple times. Therefore, the motif counting will be much more complex. Then, with such a high complexity, one may wonder how previous methods counted motifs to perform link prediction in temporal networks? The answer is that they are not scalable enough to work on the original edge streams as we are doing. They must first aggregate edges into network snapshots, which significantly decreases the number of the edges before performing motif counting (for example [2]). However, they will run into the issues of using network snapshots, which we have analyzed in the manuscript.
>
> ## Q2.  "High sensitivity of the hyperparameters"
> **We respectfully disagree with R3. Our model with a wide range of hyperparameters keeps outperforming the baselines.**
> * The optimal ranges of hyperparameters are easy to be determined as how they impact the model is very interpretable. Please check the range of each y-axis in Fig. 5 and compare it with the inductive baselines in Table 2. We may conclude that in almost the entire ranges of all three hyperparameters, our method outperforms all baselines. We zoom in the range of each y-axis and focus on the most sensitive range to provide the clearest hyperparameter analysis for the readers. As the figure shows, the only problematic regime could be when alpha, the time bias, is too large. However, in practice, it is easy to decide the proper range of \alpha as the reason why a large \alpha does not work is very clear. A too large \alpha makes the model always sample the most recent adjacent link, which loses much information and leads to bad performance. In practice, we may simply choose \alpha ~ \tau/c to avoid this issue, where \tau is the edge intensity and c is a constant from about 5 to 10. This option is uniformly good for all the datasets used in this paper.  We further clarified this point in the updated version of the paper.
>
> [1] Ashwin Paranjape, Austin R Benson, and Jure Leskovec. Motifs in temporal networks. In Proceedingsof the Tenth ACM International Conference on Web Search and Data Mining, pp. 601–610, 2017.
>
> [2] Mahmudur Rahman and Mohammad Al Hasan. Link prediction in dynamic networks using graphlet. In Joint European Conference on Machine Learning and Knowledge Discovery in Databases, pp.394–409. Springer, 2016.

---

### Author Response · Authors · 2020-11-17
**Summary of the reviewers and our response**

## Summary of reviews
We thank the reviewers for their time and valuable feedback for improving the manuscript. All the reviewers agreed that our causal anonymous walk method is a novel and effective approach to represent dynamics of temporal networks. All the reviewers also appreciated the extensive evaluation of our method by comparing with a wide range of baselines and providing complete ablation study to understand how different components contribute to the model. The reviewers also make several valuable and actionable suggestions which we will address in the revised version of the paper. We summarize the main concerns as the follows.

**Regarding the complexity**, R4 would like to see the evaluation of the complexity of our approach. R3 has questions about the benefit of our complexity over the motif counting approach. R2 is confused about the significance of our link sampling strategy as opposed to the sampling strategy adopted by TGAT.

To address these concerns, we provide the evaluation on complexity in Sec. 5.4 in the revised manuscript. One quick answer to these questions is that the complexity of our CAW-N is O(|E|), or say \beta E, where |E| is the number of edges and \beta is some constant. The constant \beta is determined by the number of walks M, the length of a walk m, the edge intensity \tau and the time bias \alpha, which finally results in only a small constant \beta. In comparison, TGAT  needs to record the entire history and has time complexity O(|E|^2). The motif counting, as it depends on tracking different combinations of edges and solving graph isomorphism testing, has complexity in general poly(|E|), where the power order decides by the size of the motifs. Overall, our approach has low memory and time complexity, which is particularly good for long edge streams. In the revised manuscript, we provided Sec. 5.4 to evaluate the complexity of our approach. We keep detailed explanation of this point in the following response to the reviewers 2, 3 and 4.

**Regarding methodology**, R1 wants to learn more about the significance of CAW as opposed to the previous AW approach. R2 would like to be clearer about the benefit of random-walk approaches and the notion of inductiveness. We will explain this point in detail in the following response to R1 and R2.

There are some other minor concerns raised by different reviewers. We directly respond to these minor concerns to the corresponding reviewers respectively.

---

### Decision · Program_Chairs · 2021-01-08
**Final Decision**

**Decision:**

Accept (Poster)

**Comment:**

The paper introduces a new method for encoding dynamics of temporal networks.  The approach, while not ground-breaking, is interesting and the results are fairly convincing.

The submission raised a number of concerns from the reviewers. They questioned the complexity of the proposed approach (R3 and R4), the clarity/readability (R2 and R1), and appropriateness of the link sampling strategy (R2), as well as raised several more minor (from my perspective) issues. I believe that the authors adequately addressed most of these concerns in their rebuttal and the revision.  R2 has confirmed that they read the rebuttal and raised their score to strong accept. Unfortunately, the other reviewers have not engaged during the discussion period, and it is unclear if they are satisfied with the clarifications and changes. Nevertheless, after reading the authors' responses and skimming through the manuscript, I believe that most concerns have been addressed, and this is a good paper that deserves to be accepted. That being said, the issue of readability has been raised by the reviewers, and, while I do not think the paper is unreadable, I do agree that there is much room for improvement. I would encourage the authors to polish the manuscript for the camera-ready version, as well as try to address the remaining concerns raised by the reviewers.